# Evacuation Strategy Considering Path Capacity and Risk Level for Cruise Ship

**Yang Liu** [1], **Huajun Zhang** [1,*], **Yu Zhan** [1], **Kunxiang Deng** [1] and **Liangzhi Dong** [2]

1　School of Automation, Wuhan University of Technology, Wuhan 430070, China; wxily@whut.edu.cn (Y.L.); 307332@whut.edu.cn (Y.Z.); 307284@whut.edu.cn (K.D.)
2　CSSC Cruise Technology Development Co., Ltd., Shanghai 200023, China; dongliangzhi@chinacruise.net.cn
*　Correspondence: zhanghj@whut.edu.cn

**Abstract:** Cruise ships are large and complex, and it is difficult to manually make a plan to evacuate people to safe areas in a short time. Evacuation time and personnel safety are both important for emergency evacuation. This paper proposes an evacuation strategy that considers the path capacity and risk level to guide evacuees in fire; it not only ensures the safety of people on dangerous paths but also reduces road congestion to shorten evacuation time. High crowd density means slow moving speed, an exponential function including straight path and stairs speed characteristics is proposed to illustrate the relationship between crowd density and moving speed. Path capacity constraints are used to avoid the congestion caused by the evacuees in a panic. In order to evacuate the evacuees in the risk areas as soon as possible, this paper divides the path into three risk levels according to carbon monoxide concentration, visibility, and temperature along the paths. The people on the higher-risk paths are given higher priority to enter evacuation paths than those on lower risk. The priority strategy evacuates the people on risk paths to safe areas in less time. This paper models the evacuation network topology of a cruise ship and simulates the evacuation process of some situations that have different numbers of evacuees and path capacity constraints. The evacuation strategies and simulation results are guidelines for the crews to guide the people to evacuate to safe areas when there is a fire accident on the cruise ship.

**Keywords:** crowd density; fire simulations; emergency evacuation; capacity constraints

## 1. Introduction

With the vigorous development of marine tourism construction, large cruise ships that can accommodate more people are the focus of today's construction [1,2]. These large cruise ships have the characteristics of complex internal structure, narrow passages, many pedestrians and few exits. When emergencies occur, pedestrians can easily follow each other during evacuation, which will lead to path congestion and even stampede accidents. Moreover, as the structure of cruise ships has become more complex, emergency evacuation strategies have become quite difficult. Therefore, in response to these challenges, emergency evacuation strategies should be continuously perfected [3–5].

In order to safely and quickly evacuate all evacuees, researchers have begun to study evacuation routes. Some scholars [6–8] have improved the Dijkstra algorithm in combination with the environmental factors that affect the evacuees, and obtained the best evacuation route. Ramesh et al. [9] used the hierarchical path planning algorithm (HIPLA) to solve the risk minimization problem in dynamic hazardous environments. Liu et al. [10] proposed an improved ant colony system (IACS) to solve the evacuation route planning problem for cruise ships. In addition to the study of evacuation routes, these scholars also involved some potential factors that affect the evacuation speed. However, current general heuristics only compute the best evacuation route from the source to the nearest exit, without considering path capacity constraints and traffic from other sources. This makes

it impossible to formulate an effective plan for evacuating a large number of people and a complex route network. Therefore, new approaches are needed to address the problem about the capacity constraints of evacuation networks.

In the early days, Lu et al. [11] proposed two heuristic algorithms, namely Single-Route Capacity Constrained Planner and Multiple-Route Capacity Constrained Planner to incorporate capacity constraints of the routes. These two algorithms are the optimal evacuation paths after considering the path capacity constraints, which are very similar to the algorithms for solving the maximum flow problem. However, these algorithms do not give personnel evacuation behavior planning, and personnel evacuation behavior planning and route planning can be combined for evacuation. In the current research, there are many studies on the strategy of considering the capacity and load of the road in the control of traffic flow. T.Q. Tang et al. [12] develop a macro traffic flow model taking into consideration road capacity to study the impact of the road capacity on traffic flow based on existing traffic flow models. Hongyan et al. [13] consider node idle capacity for maintenance analysis of transportation networks by the traffic transfer principle. Referring to the research on capacity constraints in the field of transportation, the method of the path capacity constraints is introduced into the emergency evacuation of cruise ships. For the emergency evacuation research of sudden fire, most scholars simulate the whole process of fire through PyroSim software and obtain the data of fire products in all areas [14,15]. This paper uses the method concerning dividing the risk level of path according to the concentration of fire products, and on the basis of path capacity constraints, the priority strategy of evacuation is added to speed up the evacuation of evacuees in dangerous areas.

Research on path planning algorithms is generally based on various evacuation models. The evacuation model can be divided into macro model, micro model, and mesoscopic model according to the modeling principle and details design of the personnel model [16]. At the micro level, continuous or discrete models can be used for simulation, focusing on pedestrian evacuation and spatial innovation. The movement of pedestrians is determined by factors such as pedestrian characteristics and surrounding environment. For example, Gipps et al. [17] proposed a cost-effective cell-based model. The pedestrian cellular automata model was proposed by Blue et al. [18,19]. Helbing et al. [20–22] proposed to convert the pedestrian motion model into two types of social force models. The queuing network model [23,24] is mainly used to simulate the evacuation in public places. The macro level focuses on studying the characteristics of the system from a global perspective, mainly describing the movement of pedestrian flows from the "speed–flow–density" relationship [25]. Combined with the evacuation models, it can analyze the rationality of the existing evacuation regulations and the internal structure of the buildings and provides opinions to the building designer. Compared with the macro model, the mesoscopic model contains both the characteristics of pedestrian flow and detailed individual movement, behavior, and interaction [26]. Based on the actual situation of emergency evacuation and the reference of mesoscopic model, this paper designs a priority evacuation model based on the path capacity constraints.

This paper is organized as follows: Section 2 introduces the method of establishing the network topology of the cruise ship and fire simulation. Section 3 expounds the design of the evacuation strategy, including the capacity constraints of the path and the evacuation priority according to the risk level of the path. The simulation results and analysis are in Section 4. Section 5 summaries the conclusions.

## 2. Indoor Environment Modeling

In order to verify the evacuation model in this paper and analyze the advantages and disadvantages of the model performance, it is necessary to investigate and obtain evacuation scene data and build an evacuation network topology for the evacuation scene. The evacuation environment used in this paper is the three-tier cruise ship given in the European "SAFEGUARD" project [27]. The number, type, speed, and evacuation strategy of pedestrians in different areas are set in the evacuation environment. Based on the

evacuation environment of the cruise ship, this paper verifies the rationality of the model through the simulation of the evacuation. Based on the data of the cabin, stairs, passage, bar area, assembly area, and storage area of the cruise ship, this research completes the construction of the cruise ship model. The construction diagram of the first layer is shown in Figure 1.

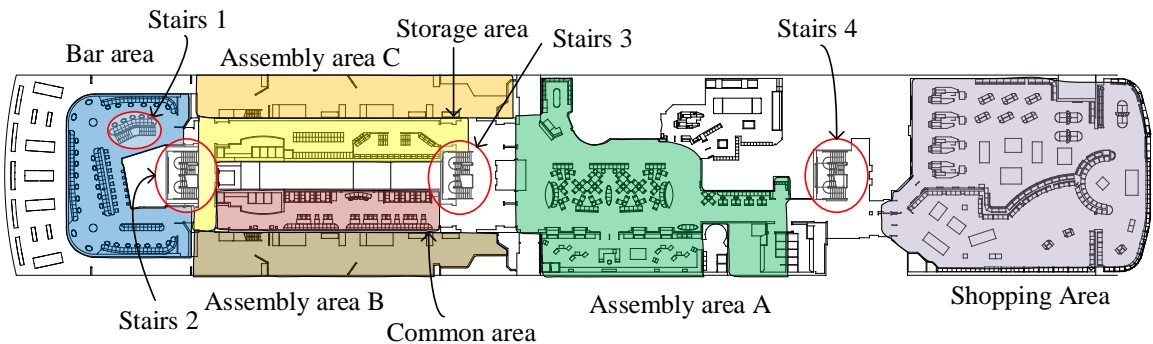

**Figure 1.** The construction diagram of the first layer.

### 2.1. Networking of the Cruise Ship Model

In order to meet the application environment of the evacuation model in this paper, the two-dimensional cruise ship model needs to be converted into an evacuation network topology. After investigating the ways of transforming network topology of large-scale buildings, the steps for constructing an evacuation network topology are as follows:

1. Read cruise ship data;
2. Set up nodes for the centers of different areas in the cruise ship;
3. Judge whether different areas are connected to each other;
4. Connect the central points of the interconnected areas.

According to the functions of different facilities in the cruise ship, the cruise ship can be divided into bars, restaurants, stairs, shopping areas, cabins, storage areas, passages, decks, and other areas. These areas can be connected by passages or doors. Therefore, this method first selects the center points of these areas, and then judges whether there are doors or passages between different areas to determine the connectivity between the areas. Finally, doors and passages are used to connect central points and then an evacuation network topology is generated. In addition, in order to study the impact of path capacity constraints and evacuation priority on evacuation, it is necessary to set up the fire location and the exit as the prerequisite for evacuation simulation. In the "SAFEGUARD" project, pedestrians were designated to flee to assembly area C. Assembly area C contains the rescue materials needed for emergency evacuation, so this paper takes the node leading to assembly area C as the exit. The materials in the ship are generally non-flammable, and the place where a fire occurs is often the luggage of tourists. Therefore, this paper uses the storage area as the place where the fire broke out.

### 2.2. Description of Evacuation Network Topology Attributes

In this paper, the evacuation network topology is represented by $G(V, E, W)$, where $V$ represents the set of nodes in the evacuation network topology, $E$ represents the set of paths, $W$ represents the collection of the length and width of each side. This paper defines $i$ as the first node of the path and $j$ as the end node of the path. $w_{ij} = (a_{ij}, b_{ij})$ represents the length and width of each path. $a_{ij}$ represents the length of the path, and its value is the distance between the first node and end node. $b_{ij}$ represents the width of the path, and its value is the shortest distance between the sides of the path. Assuming that there are $n$ nodes in total, there is $i, j = 0, 1, 2, \cdots, n$.

If the evacuation network topology is to be applied to the evacuation model of this paper, the attributes defined by the above nodes and paths are not enough to meet the

requirements. In order to use the priority evacuation model in this paper to simulate the evacuation on the evacuation network topology, this paper expands the attributes of nodes and paths. The detailed descriptions are shown in Tables 1 and 2.

**Table 1.** Attribute description and scope of the node.

| Symbol | Illustrate | Scope |
|:---:|:---:|:---:|
| $i$ | node number | $i \in \{0, 1, \cdots, n-1\}$ |
| $x_i$ | abscissa | $x_i \in R$ |
| $y_i$ | ordinate | $y_i \in R$ |
| $n_i$ | adjacent node set | $n_i \subsetneq \{0, 1, \cdots, n-1\}$ |
| $p_i$ | adjacent path set | $p_i \subsetneq \{0, 1, \cdots, m-1\}$ |
| $SR_i$ | the shortest route to the exit | $SR_i \subsetneq \{0, 1, \cdots, n-1\}$ |
| $SRL_i$ | the shortest route length to the exit | $SRL_i \in [0, +\infty)$ |

**Table 2.** Attribute description and scope of the path.

| Symbol | Illustrate | Scope |
|:---:|:---:|:---:|
| $e_{ij}$ | path number | $e_{ij} \in \{0, 1, \cdots, m-1\}$ |
| $i$ | first node number | $i \in \{0, 1, \cdots, n-1\}$ |
| $j$ | end node number | $j \in \{0, 1, \cdots, n-1\}$ |
| $a_{ij}$ | length | $a_{ij} \in (0, +\infty)$ |
| $b_{ij}$ | width | $b_{ij} \in (0, +\infty)$ |
| $s_{ij}$ | area | $s_{ij} \in (0, +\infty)$ |
| $N_{ij}$ | number of people | $N_{ij} \in (0, +\infty)$ |
| $danger_{ij}$ | risk level | $danger_{ij} \in \{0, 1, 2\}$ |
| $D_{ij}$ | crowd density | $D_{ij} \in (0, +\infty)$ |

In the description of node attributes 1, *n* represents the total number of nodes; m represents the total number of paths. In the description of the path attributes in Table 2, the calculation equation of area $s_{ij}$ and the crowd density $D_{ij}$ on the path are as shown in Equations (1) and (2). In addition, $danger_{ij} = 0$ indicates that the path has the highest risk level. $danger_{ij} = 1$ indicates that the path has lower risk level. $danger_{ij} = 2$ means the path is safe.

$$s_{ij} = a_{ij}b_{ij} \tag{1}$$

$$D_{ij} = \frac{N_{ij}}{s_{ij}} \tag{2}$$

In Equation (2), $N_{ij}$ represents the total number of people on the path $e_{ij}$, and $N_{ij}$ is determined by the location attributes of all evacuees at each moment. Through looking up the literature [28], it is found that the speed of evacuation of people is closely related to the crowd density of the path. The setting of the attribute $D_{ij}$ is to provide the moving speed of people during evacuation.

This paper focuses on the impact of setting path capacity constraints and evacuation priority on the evacuation results. The method of determining the risk level of the path can be simplified. The judgment of risk level of the path can be based on the value of visibility, CO concentration, and temperature. Zhu et al. [29] studied the impact of these indicators on evacuation. In the actual evacuation, the temperature sensor, CO sensor, and visibility sensor can be used to detect the fire situation of each path. In addition, the sprinkler and smoke exhaust system can reduce and control a certain degree of fire. The influence of these three indicators on the evacuation coefficient [29] is shown in Equations (3)–(5).

$$f_m = \min(1, \max(0.2, 1 - 0.34(3 - VIS))) \tag{3}$$

$$f_c = \begin{cases} 1 & (\phi(CO) < 0.1) \\ 1 - [0.2125 + 1.788\phi(CO)] \cdot \phi(CO)t & (0.1 \le \phi(CO) < 0.25) \\ 0 & (\phi(CO) \ge 0.25) \end{cases} \quad (4)$$

$$f(\theta_s) = \begin{cases} 1 & (\theta_s \le \theta_{cr1}) \\ \dfrac{(v_{max} - 1.2)\left(\frac{\theta_s - \theta_{cr1}}{\theta_{cr2} - \theta_{cr1}}\right)^2}{v_o} + 1 & (\theta_{cr1} < \theta_s \le \theta_{cr2}) \\ \dfrac{v_{max}}{1.2}\left[1 - \left(\frac{\theta_s - \theta_{cr2}}{\theta_d - \theta_{cr2}}\right)^2\right] & (\theta_{cr2} < \theta_s \le \theta_d) \end{cases} \quad (5)$$

Equation (3): $VIS$ is the visibility of the evacuees, meter; Equation (4): $\phi(CO)$ is the volume fraction of $CO$, %; $t$ is the evacuee' exposure time, min. In Equation (5): $\theta_s$ is the actual temperature of the fire site, °C; $v_{max}$ is the maximum escape speed, m/s; $\theta_{cr1}$ is the temperature at which personnel feel uncomfortable, 30 °C in the paper; $\theta_{cr2}$ is the temperature at which personnel are injured, 60 °C in the paper; $\theta_d$ represents the temperature that caused serious injury to personnel, 120 °C in the paper. The smaller $f_m$, $f_c$, $f(\theta_s)$, the greater the impact on the evacuee' moving speed. According to the range divided by Equations (3)–(5) corresponding to the three indicators of degrees of influence on evacuation, three risk levels of the path are defined according to the range, as shown in Figure 2.

---

**Get the risk level of all paths**

**initialization:** Set $danger_{ij}$ to 0

**input** VIS, $\phi(CO)$, $\theta_s$

**for** each path in Graph

    If $0 \le VIS < 0.6 \text{ or } 0.25 \le \phi(CO) \text{ or } 60 \le \theta_s < 120$:

        $danger_{ij} = 0$

    else if $0.6 \le VIS < 3 \text{ or } 0.1 \le \phi(CO) < 0.25 \text{ or } 30 \le \theta_s < 60$:

        $danger_{ij} = 1$

    else if $3 \le VIS \text{ or } \phi(CO) < 0.1 \text{ or } 0 \le \theta_s < 30$:

        $danger_{ij} = 2$

**return** risk level of all paths

---

**Figure 2.** Pseudocode for judging path risk level.

In Figure 2, $danger_{ij} = 2$ means that the risk level of the path is low, which is a safe state; $danger_{ij} = 1$ means that the risk level of the path is relatively high, which is a relatively dangerous state; $danger_{ij} = 0$ means that the risk level of the path is high, which is a dangerous state. When constructing the network topology for some other large buildings, the extended attributes of this paper can be used as a reference, such as crowd density, the shortest route, the types of nodes, and the risk level of paths. These extended attributes are conducive to the research of evacuation algorithms and can be applied to large buildings with similar evacuation structures [30].

*2.3. Cruise Ship Data and Evacuation Network Topology*

According to the above definition and through the PyroSim software, the fire numerical analysis is carried out, and three indicators are monitored based on each node of the network topology. The first-tier cruise evacuation network topology constructed is shown in Figure 3, which is an evacuation network topology formed by connecting the central points of different regions. Since the fire area is generally in the storage area, the storage area corresponds to node 6. Therefore, this paper assumes that node 6 is the fire node. In addition, node 14 is connected to the aggregation area, and it is assumed that node 14 is an exit node. The fire simulation data is obtained by detecting the fire product index of each node in the network topology, and the risk level of the path at different times is defined as

the highest risk level of the nodes at both nodes of the path, as shown in Figure 4. Table 3 includes the moment of all nodes when the risk level changes over time.

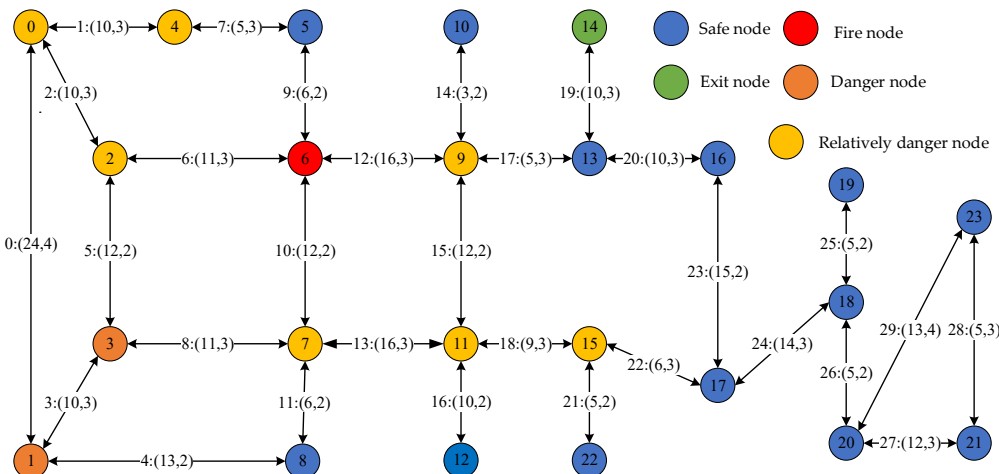

**Figure 3.** Cruise ship evacuation network topology of the 200th second of the fire.

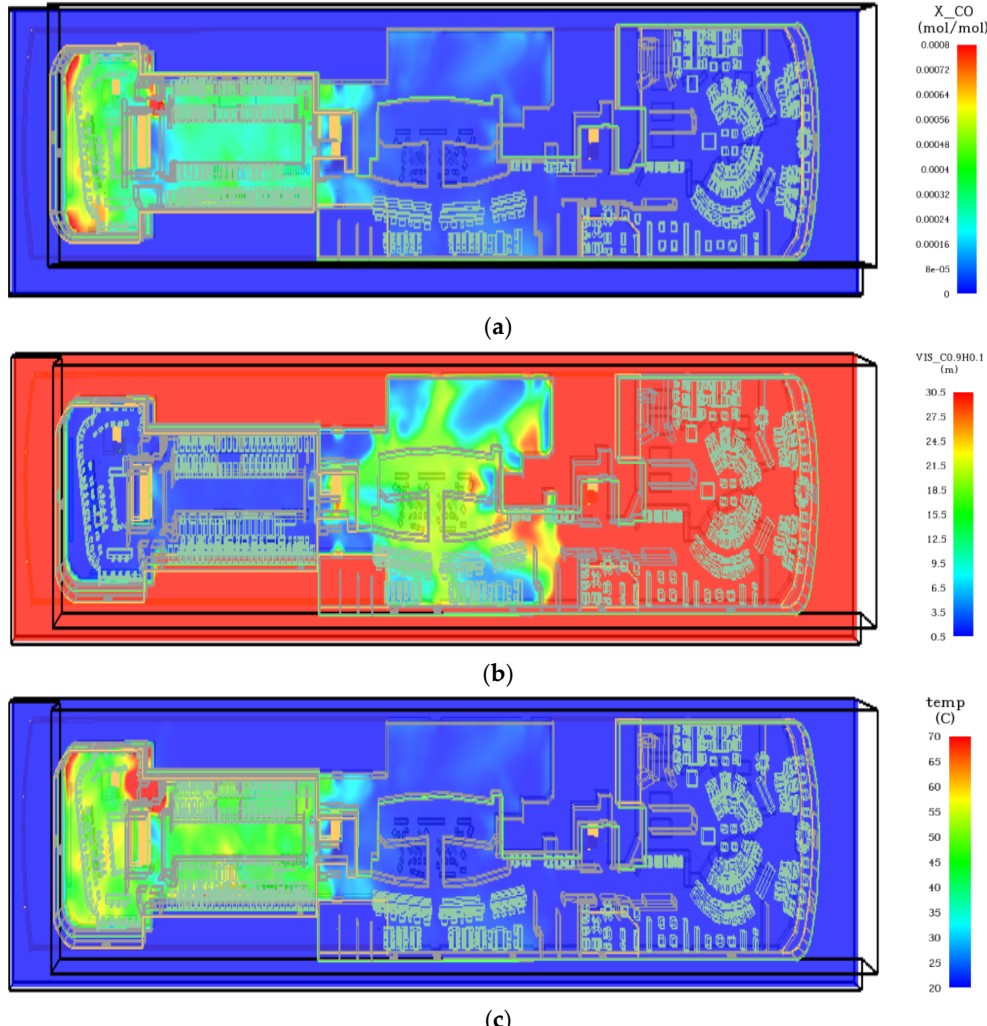

**Figure 4.** The heat map of CO concentration, visibility, and temperature corresponding to the fire at the 200th second. (**a**) The heat map of CO concentration; (**b**) The heat map of visibility; (**c**) The heat map of temperature.

**Table 3.** Dynamic change data of the cruise ship risk level under fire situation.

| *i* | T1 (s) | T2 (s) | *i* | T1 (s) | T2 (s) |
|---|---|---|---|---|---|
| 0 | 109 | 201 | 12 | None | None |
| 1 | 110 | 175 | 13 | 229 | None |
| 2 | 115 | 204 | 14 | None | None |
| 3 | 123 | 198 | 15 | 172 | None |
| 4 | 118 | 218 | 16 | 233 | None |
| 5 | None | None | 17 | 262 | None |
| 6 | 21 | 51 | 18 | None | None |
| 7 | 115 | None | 19 | None | None |
| 8 | None | None | 20 | None | None |
| 9 | 144 | None | 21 | None | None |
| 10 | None | None | 22 | None | None |
| 11 | 151 | None | 23 | None | None |

In Table 3, T1 represents the moment when the path risk level becomes relatively dangerous, and T2 represents the moment when the path risk level becomes dangerous. None indicates that each node cannot reach a relatively dangerous state or a dangerous state during the entire fire process. The data in Table 3 can be used to determine the path risk level corresponding to all moments after the fire occurs.

## 3. Evacuation Model

In the event of fire and other emergencies, the evacuation model can simulate the impact of different evacuation strategies on the evacuation results. Evaluation of evacuation results and evacuation models can use evacuation time as the indicator. There are two types of evacuation time, available safe egress time (*ASET*) and required safe egress time (*RSET*) [31,32]. The different types of times included in *ASET* and *RSET* are shown in Equations (6) and (7).

$$ASET = t_{start} + t_{evac} + t_{marg} \qquad (6)$$

$$RSET = t_{start} + t_{evac} \qquad (7)$$

In Equations (6) and (7), $t_{start}$ represents the time required for the pedestrian to detect the fire, call the police, and then final response. $t_{evac}$ represents the time required for the actual movement of the pedestrian. $t_{marg}$ represents the time for the building to reach its fire-resistance limitation minus *RSET* [33]. Only when the *ASET* is greater than *RSET* can the safety of pedestrians be guaranteed. In order to accurately estimate RSET when simulating pedestrians' evacuation, evacuees' moving speed is an important indicator. Evacuees' moving speed is closely related to the crowd density on the path.

### 3.1. Evacuees' Moving Speed

After completing the construction of the evacuation network topology, it is also necessary to obtain the moving speed of the evacuees when evacuating in different locations. When the evacuation speed is closer to the actual evacuation speed, the evacuation time will be more accurate by simulation. Most of the speed models studied by different scholars are based on crowd density. After literature research [34–38], the literature review on the relationship between evacuees' moving speed and crowd density was summarized and analyzed, as shown in Figures 5–7. They correspond to the relationship between the crowd density and the speed of the evacuees on the straight path, ascending and descending stairs.

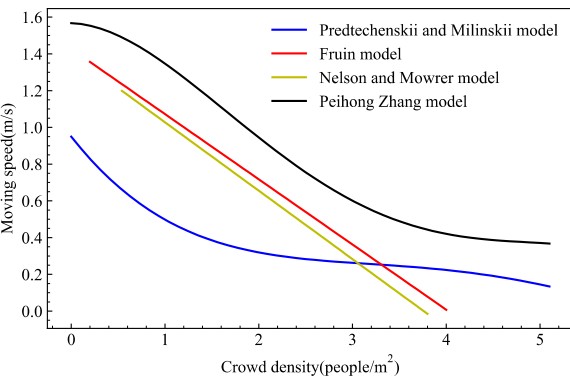

**Figure 5.** The evacuees' speed models of moving on straight paths of various scholars.

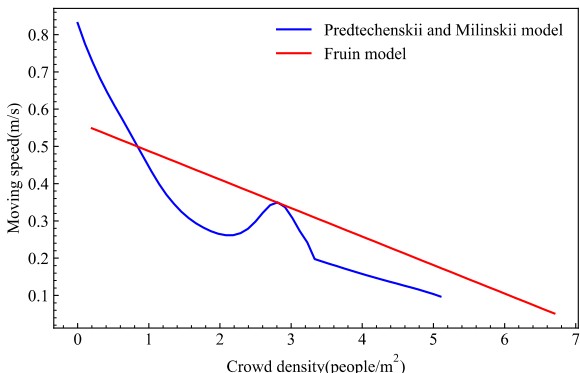

**Figure 6.** The evacuees' speed model of moving up the stairs of various scholars.

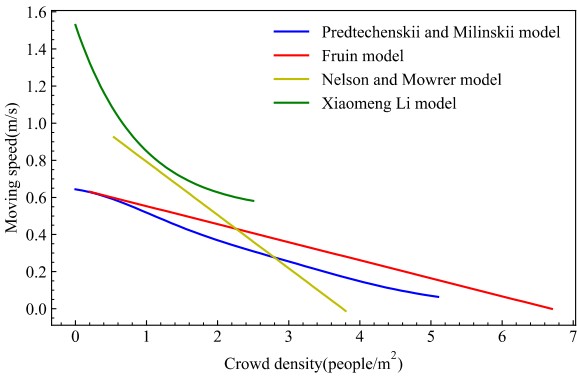

**Figure 7.** The evacuees' speed model of descending stairs of various scholars.

Figures 5–7 show the evacuees' moving speed on the stairs and the straight path, respectively. That the speed models proposed by scholars are based on the evacuation experimental data in different buildings has led to large differences among their models. It can be seen from Figure 4 that, when the crowd density exceeds 4 people/m$^2$, the Fruin model and the Nelson–Mowrer model will no longer be applicable. In practice, the high crowd density will not cause evacuees' moving speed to be 0. Moreover, when the crowd density is close to 0 people/m$^2$, there is a big difference among the scholar models. Therefore, scholars' speed models need to be improved. In order to reduce the influence of the differences and improve the evacuees' speed model, this paper integrates the speed models of various scholars to make the new model closer to the actual situation.

In order to integrate these pedestrian speed models, this paper first selected the data points that need to be fitted. The fitting steps are as follows:

1. Start from D = 0.1 people/m$^2$, use 0.1 people/m$^2$ as the step length to select value of evacuees' moving speed of each scholar speed model, finally generate multiple sets of speed data for each research scholar;
2. Calculate the average of the moving speed sets of scholars in step (1);
3. Use an exponential function to fit these data and obtain the relationship between evacuees' moving speed and crowd density on straight paths, ascending stairs, and descending stairs, respectively.

In step (3), the method of fitting evacuees' moving speed model through the exponential function can be converted into a linear programming problem. The least square method is used to solve the optimal solution. The sample point obtained in steps (1) and (2) is $(x_i, y_i)$, $i = 1, 2, \cdots, n$, and the fitting curve is set as:

$$y = ae^{bx} \tag{8}$$

Take the logarithm of the fitted curve and the fitted value is:

$$\ln y = \ln a + bx \tag{9}$$

$$\ln \hat{y}_i = \ln a + bx_i \tag{10}$$

From Equations (9) and (10), the loss function of the curve can be obtained as:

$$L = \sum_{i=1}^{n} (\ln y_i - \ln \hat{y}_i)^2 = \sum_{i=1}^{n} (\ln y_i - bx_i - \ln a)^2 \tag{11}$$

It can be seen from the above equations that the objective function of the linear programming model is:

$$\min L = \min \left[ \sum_{i=1}^{n} (\ln y_i - bx_i - \ln a)^2 \right] \tag{12}$$

This paper uses the least square method to solve the optimal solution of the linear programming model, as shown in Equations (14) and (15).

$$\begin{cases} \frac{\partial L}{\partial b} = -2 \sum_{i=1}^{n} x_i (\ln y_i - bx_i - \ln a) = 0 \\ \frac{\partial L}{\partial \ln a} = -2 \sum_{i=1}^{n} (\ln y_i - bx_i - \ln a) = 0 \end{cases} \tag{13}$$

$$\hat{b} = \frac{n \sum_{i=1}^{n} x_i \ln y_i - \sum_{i=1}^{n} \ln y_i \sum_{i=1}^{n} x_i}{n \sum_{i=1}^{n} x_i^2 - \sum_{i=1}^{n} x_i \sum_{i=1}^{n} x_i}, \quad \ln \hat{a} = \frac{\sum_{i=1}^{n} x_i^2 \sum_{i=1}^{n} \ln y_i - \sum_{i=1}^{n} x_i \sum_{i=1}^{n} x_i \ln y_i}{n \sum_{i=1}^{n} x_i^2 - \sum_{i=1}^{n} x_i \sum_{i=1}^{n} x_i} \tag{14}$$

Finally, the fitting result is shown in Equation (15).

$$\begin{cases} v_L = 1.725 e^{-0.4791D} \\ v_U = 0.737 e^{-0.3437D} \\ v_D = 1.161 e^{-0.4591D} \end{cases} \tag{15}$$

In the Equation (15), $v_L$, $v_U$, and $v_D$ correspond to the evacuees' moving speed on straight paths, ascending stairs, and descending stairs, respectively. D represents the crowd density. The fitted evacuees' moving speed curve can well reflect the negative correlation between the evacuees' moving speed and the crowd density. This method solves the problem that most scholars have large differences and inconsistencies in the evacuees' moving speed, as shown in Figures 8–10.

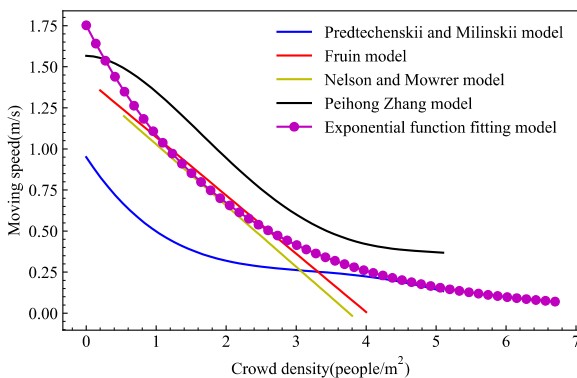

**Figure 8.** The fitting model of evacuees' moving speed on straight paths.

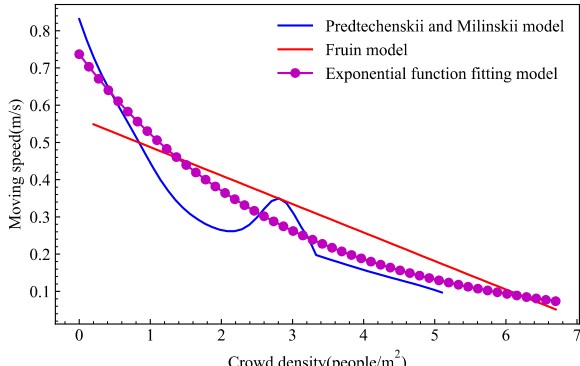

**Figure 9.** The fitting model of the moving speed of evacuees ascending stairs.

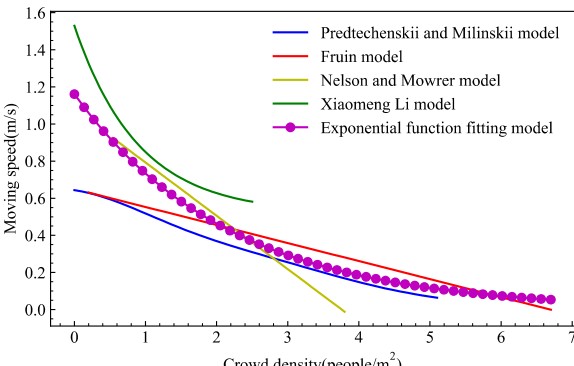

**Figure 10.** The fitting speed model of evacuees descending stairs.

In the simulation of dynamic evacuation, the values corresponding to the attributes of all evacuees need to be continuously updated. The total number of evacuees in each path also needs to be updated. To this end, it is necessary to first define the attributes and scopes on the evacuee, as shown in Table 4.

**Table 4.** Evacuee's attributes and scope.

| Symbol | Illustrate | Scope |
|---|---|---|
| $x$ | evacuee's serial number | $x \in \{0, 1, 2, \cdots, N-1\}$ |
| $x_{ij}$ | serial number of the path | $x_{ij} \in \{0, 1, \cdots, m-1\}$ |
| $x.i$ | first node of the path | $x.i \in \{0, 1, \cdots, n-1\}$ |
| $x.j$ | end node of the path | $x.j \in \{0, 1, \cdots, n-1\}$ |
| $v_x$ | moving speed | $v_x = 1.725e^{-0.4791D_{ij}}$ |
| $length_i$ | the distance between the evacuee and the node $i$ | $length_i \in [0, a_{ij}]$ |
| $order_x$ | evacuee's movement flag | $order_x \in \{0, 1\}$ |
| $priority_x$ | evacuation priority | $priority_x = \{0, 1, 2\}$ |
| $x.np$ | serial number of evacuee's next path to go | $x.np \in \{0, 1, \cdots, m-1\}$ |
| $x.nn$ | serial number of evacuee's next node to go | $x.nn \in \{0, 1, \cdots, n-1\}$ |
| $route_x$ | evacuee's evacuation route | $route_x \subsetneq \{0, 1, \cdots, n-1\}$ |

In Table 4, $N$ represents the total number of all evacuees. When $order_x = 0$, the evacuee stops moving forward. When $order_x = 1$, the evacuee continues to move forward. The value of $priority_x$ is determined by the risk level of the path. When the risk level of the path to which the person belongs is 0, that is $danger_{ij} = 0$, then $priority_x = 0$ is established. This means that the evacuation priority is the highest. By analogy, $priority_x = 1$ means the evacuation priority is lower, and $priority_x = 2$ means the evacuation priority is the lowest.

*3.2. Priority Evacuation Model with Path Capacity Constraints and Fire Situation*

The priority evacuation model is proposed to take into account the different degrees of safety of different paths. In addition, when people are evacuated, many evacuees on different paths may flood into one path. The crowd density at this intersection will soon reach a high value, which will lead to serious path congestion and even crowd trampling accidents in actual situations. Based on this phenomenon, the priority evacuation model was established on the basis of Table 4. The capacity constraints of the path can be converted into the maximum crowd density setting of the path, see Equation (16).

$$D_{\max} = \frac{N_{\max}}{s_{ij}} \tag{16}$$

$N_{\max}$ represents the maximum evacuees' capacity on the path. $D_{\max}$ represents the maximum crowd density of the path. By Equations (16), the constraints on total number of evacuees on all paths can be converted into an equal maximum crowd density $D_{\max}$. In addition, the detailed steps of the priority evacuation model are shown in the flow chart in Figure 11.

The upper part of Figure 11 shows the determination of $order_x$ by the evacuation priority. $length_{x.nn} \geq dis$ means that the distance between the evacuee and the next node to go to is greater than $dis$. $dis$ represents the threshold for evacuees to stay and wait. Only when the distance from the evacuees to the next node is less than this value will the evacuation priorities need to be compared between evacuees; otherwise, they keep moving forward. This paper takes 2 m as a hypothesis to complete the subsequent evacuation simulation experiments. The lower part shows the movement model of the evacuee evacuation, which is based on the speed–density relationship, updates the attributes of all evacuees in real time. t indicates the program update time.

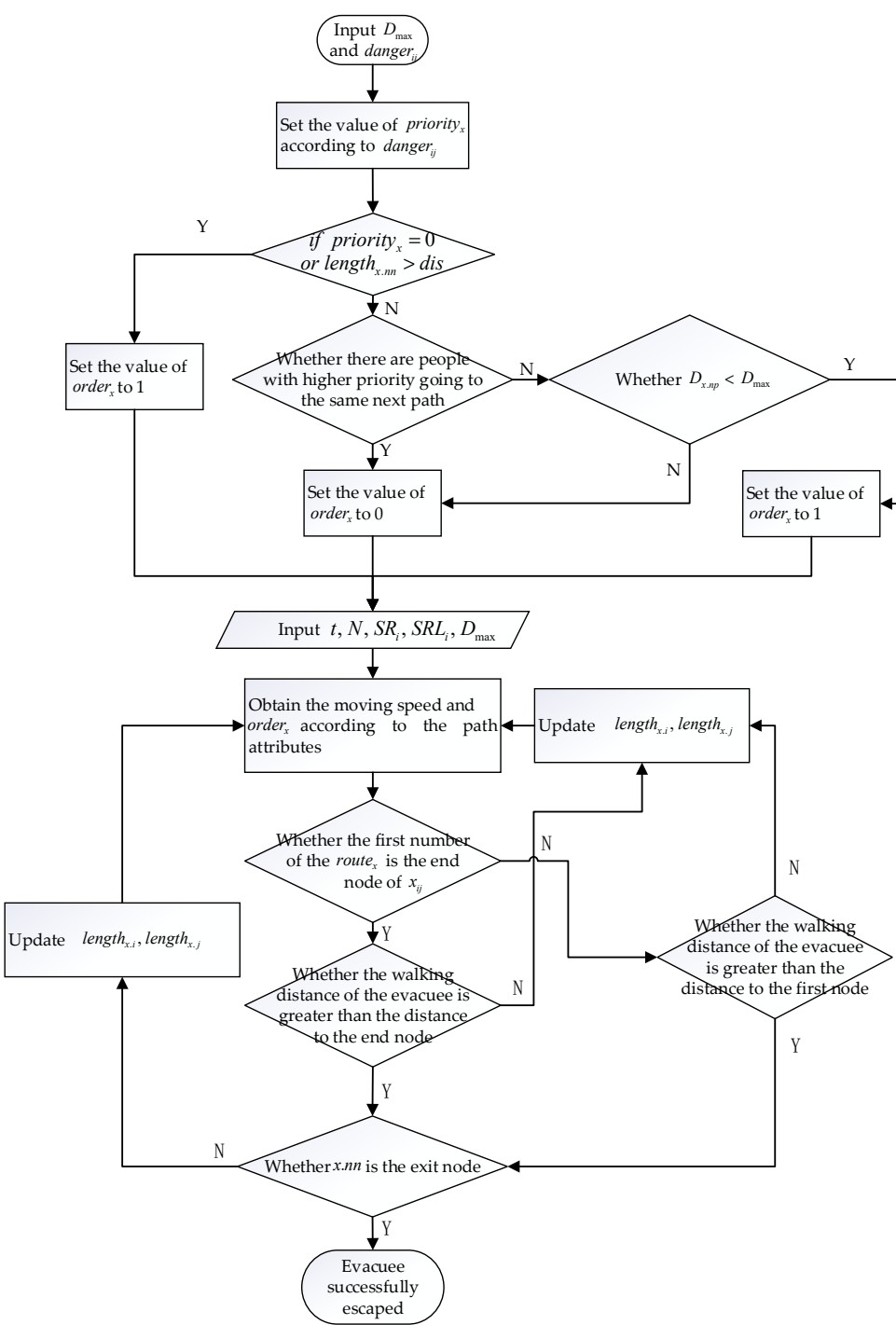

**Figure 11.** Flow chart for priority evacuation model.

### 3.3. Evacuation Route Strategy

The evacuation simulation in Section 4 of this paper focuses on embodying the advantages of the priority evacuation model. The dynamic path planning algorithm will be developed in the future. This paper provides the evacuation strategy of evacuees on different paths by still using the traditional path planning algorithm to find the shortest route from the exit. This route does not include the paths connected to the fire area. The Dijkstra algorithm is a typical algorithm for solving the shortest route of a single source point in a weighted graph. It adopts the greedy idea and continuously expands from the source point to the connected nodes [39,40].

The shortest route from all nodes to the exit is shown in Table 5. All evacuees will be evacuated according to the route in Table 5.

**Table 5.** The shortest route from each node to the exit node.

| Start-End | Distance | Route |
|---|---|---|
| 0–14 | 53 | {0, 2, 6, 9, 13, 14} |
| 1–14 | 63 | {1, 8, 7, 11, 9, 13, 14} |
| 2–14 | 43 | {2, 6, 9, 13, 14} |
| 3–14 | 55 | {3, 7, 11, 9, 13, 14} |
| 4–14 | 43 | {4, 5, 6, 9, 13, 14} |
| 5–14 | 38 | {5, 6, 9, 13, 14} |
| 6–14 | 32 | {6, 9, 13, 14} |
| 7–14 | 44 | {7, 11, 9, 13, 14} |
| 8–14 | 50 | {8, 7, 11, 9, 13, 14} |
| 9–14 | 16 | {9, 13, 14} |
| 10–14 | 19 | {10, 9, 13, 14} |
| 11–14 | 28 | {11, 9, 13, 14} |
| 12–14 | 38 | {12, 11, 9, 13, 14} |
| 13–14 | 11 | {13, 14} |
| 14–14 | 0 | {14} |
| 15–14 | 37 | {15, 11, 9, 13, 14} |
| 16–14 | 21 | {16, 13, 14} |
| 17–14 | 36 | {17, 16, 13, 14} |
| 18–14 | 50 | {18, 17, 16, 13, 14} |
| 19–14 | 55 | {19, 18, 17, 16, 13, 14} |
| 20–14 | 55 | {20, 18, 17, 16, 13, 14} |
| 21–14 | 67 | {21, 20, 18, 17, 16, 13, 14} |
| 22–14 | 42 | {22, 15, 11, 9, 13, 14} |
| 23–14 | 68 | {23, 20, 18, 17, 16, 13, 14} |

## 4. Model Validation

In order to verify the effect of the priority evacuation model on the emergency evacuation of cruise ships, as well as the impact of the path capacity constraints on the evacuation time, this paper completes three types of evacuation models for evacuation simulation based on Python, including: 1. There is neither path capacity constraints nor the evacuation priority; 2. there are path capacity constraints but no the evacuation priority; 3. existing path capacity constraints and the evacuation have priority. The software version used in the experiment is Python 3.6 [41] and Pycharm 2019.3.3x64. The computer hardware is Intel Core i5-1035G1, clocked at 1.00 Ghz, 16 G memory. Figure 11 shows the flow chart of the evacuee's movement model over time.

The evacuation simulation is based on the evacuee's movement logic in Figure 11 to verify the impact of path capacity constraints and the evacuee's priority evacuation under fire conditions. For different evacuation models, the initial position of the evacuees remains the same. In addition, this paper combines RSET and the escaped dangerous route time (EDRT) for evacuees to evaluate the evacuation models. RSET represents the time required for all evacuees to escape the building. EDRT represents the time it takes for all evacuees on the path where the risk level will become dangerous during the fire process to escape to the safe path.

### 4.1. No Path Capacity Constraints and Evacuation Priority Model

When the path does not have capacity constraints and the evacuation priority is not added, evacuees are set to be randomly distributed on different paths of the cruise ship. As total number of evacuees increases, the evacuation results are shown in Table 6.

**Table 6.** Relationship between the number of evacuees and evacuation time.

| N | RSET (s) | EDRT (s) | N | RSET (s) | EDRT (s) |
|---|---|---|---|---|---|
| 50 | 42 | 30 | 400 | 333 | 37 |
| 100 | 46 | 30 | 450 | 430 | 39 |
| 150 | 51 | 32 | 500 | 400 | 42 |
| 200 | 63 | 32 | 550 | 1827 | 44 |
| 250 | 87 | 33 | 600 | 4599 | 47 |
| 300 | 141 | 34 | 650 | 11,197 | 50 |
| 350 | 221 | 36 | 700 | 9666 | 53 |

In Table 6, the two types of evacuation time will increase with the increase of the number of evacuees. For fires in buildings, ASET represents the time required for the building to reach the fire-resistance limitation. The fire-resistance rating is a grading scale that measures the fire resistance of a building. It is determined by the combustion performance and fire-resistance limitation of the components that make up the building. The fire-resistance rating of a building is one of the most basic measures in the fire protection technical measures specified in the building design fire protection code. The factors that affect the selection of fire-resistance levels are: The importance of the building, the nature of use and fire hazard, the height and area of the building, the size of the fire load, and other factors. The main building components of Class I fire-resistant buildings are all non-combustible. The main building components of Class II fire-resistant buildings are non-combustible except for the ceiling, which is difficult to burn. The roof load-bearing components of Class III fire-resistance buildings are flammable. The fire wall of the Class IV fire-resistance building is non-combustible, and the rest are flame retardant and flammable [42]. For civil buildings and high-rise buildings with Class I and II fire resistance, ASET generally takes 5–7 min. For civil buildings and high-rise buildings with Class III and IV fire resistance, ASET generally takes 2–4 min [42]. The materials in cruise ships are generally non-flammable, so this paper uses ASET = 7 min as a reference indicator to determine the maximum capacity within the safety limitation of different evacuation models [43]. The relationship between the total number of evacuees and the evacuation time drawn according to the data in Table 4 is shown in Figure 12.

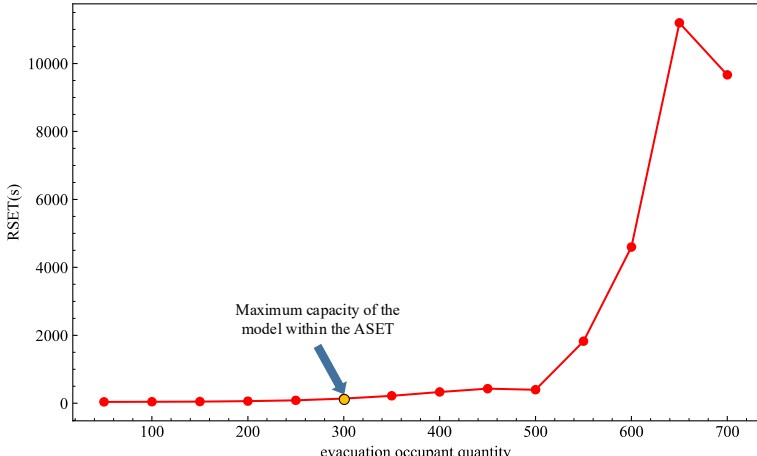

**Figure 12.** The relationship between RSET and the number of evacuees.

It can be analyzed from Figure 12 that the growth rate of RSET is not obvious before the number of evacuees reaches 500. After the number of evacuees reaches 500, the growth rate of RSET gradually accelerates. This is because when there are too many evacuees, the evacuation model does not set the path capacity constraints, which does not make full use of the free space of each path, and eventually causes serious congestion on some paths. In addition, according to the settings of ASET, it can be judged that when the number of

evacuees is less than or equal to 400, RSET is less than 7 min. When the number of evacuees is more than 400, RSET is greater than 7 min. Therefore, the authors of this paper believe that 400 evacuees is the maximum capacity of the model. This means that the model can accommodate up to 400 evacuees within the safety limitation. Moreover, the results of the optimization of the evacuation model are evaluated by using the maximum capacity within the safety limitation, RSET, EDRT, and the growth rate of RSET and EDRT.

### 4.2. Evacuation Model with Path Capacity Constraints

When there are constraints for the number of evacuees on the path, $D_{\max}$ is the maximum allowable crowd density on the path. From the flow chart of the evacuation model in Figure 11, it can be seen that, when evacuees are only *dis* away from the next path, the evacuees will start to be limited to the capacity of the next path that evacuees are going to. If the next path meets the crowd density exceeding $D_{\max}$, then the evacuees remain until the next path is allowed to pass. Literature research [44] shows that the early warning threshold of crowd density in the crowd exercise state is 4–6 people/m$^2$. In this paper, a comparative analysis of the evacuation time for $D_{\max} \in \{0.5, 1, 2, 3, 4, 5, 6\}$ (people/m$^2$) is carried out, and the results are shown in Tables 7 and 8.

**Table 7.** The relationship between the number of evacuees and RSET when $D_{\max}$ is different.

| N | 0.5 | 1 | 2 | 3 | 4 | 5 | 6 |
|---|-----|---|---|---|---|---|---|
| 50 | 42 | 42 | 42 | 42 | 42 | 42 | 42 |
| 100 | 56 | 46 | 46 | 46 | 46 | 46 | 46 |
| 150 | 73 | 56 | 51 | 51 | 51 | 51 | 51 |
| 200 | 89 | 71 | 62 | 62 | 63 | 63 | 63 |
| 250 | 117 | 87 | 72 | 72 | 72 | 77 | 85 |
| 300 | 135 | 96 | 83 | 84 | 84 | 92 | 106 |
| 350 | 154 | 127 | 95 | 95 | 97 | 108 | 121 |
| 400 | 180 | 135 | 107 | 110 | 107 | 121 | 143 |
| 450 | 187 | 152 | 124 | 127 | 130 | 124 | 168 |
| 500 | 217 | 174 | 146 | 133 | 147 | 168 | 188 |
| 550 | 231 | 175 | 153 | 151 | 201 | 187 | 213 |
| 600 | 257 | 203 | 173 | 173 | 224 | 270 | 246 |
| 650 | 303 | 226 | 187 | 200 | 277 | 256 | 330 |
| 700 | 292 | 247 | 254 | 224 | 267 | 268 | 319 |

**Table 8.** The relationship between the number of evacuees and EDRT when $D_{\max}$ is different.

| N | 0.5 | 1 | 2 | 3 | 4 | 5 | 6 |
|---|-----|---|---|---|---|---|---|
| 50 | 30 | 30 | 30 | 30 | 30 | 30 | 30 |
| 100 | 35 | 30 | 30 | 30 | 30 | 30 | 30 |
| 150 | 48 | 42 | 32 | 32 | 32 | 32 | 32 |
| 200 | 68 | 48 | 34 | 36 | 33 | 32 | 32 |
| 250 | 84 | 46 | 40 | 35 | 36 | 33 | 33 |
| 300 | 103 | 63 | 54 | 41 | 38 | 34 | 34 |
| 350 | 125 | 80 | 67 | 46 | 46 | 36 | 36 |
| 400 | 125 | 80 | 46 | 50 | 50 | 40 | 37 |
| 450 | 157 | 94 | 85 | 68 | 55 | 44 | 39 |
| 500 | 179 | 116 | 88 | 88 | 60 | 50 | 42 |
| 550 | 194 | 134 | 114 | 80 | 87 | 84 | 44 |
| 600 | 217 | 153 | 113 | 99 | 95 | 104 | 101 |
| 650 | 245 | 144 | 139 | 116 | 108 | 114 | 117 |
| 700 | 281 | 178 | 160 | 144 | 110 | 111 | 90 |

From the data in Tables 6–8, it can be seen that the two types of evacuation time in Tables 7 and 8 are the same as those in Table 6 when the number of evacuees is 50. When the number of evacuees increases, the REST in Table 7 is significantly less than the REST

in Table 6. This proves that the number of evacuees on each path has been controlled after the path capacity constraints. Furthermore, the path will not be too congested, and evacuees' moving speed and safety will be guaranteed. For the actual evacuation situation, the method of capacity constraints can be used for reference. The evacuees can be equipped with smart bracelets as signs. Crews who direct the evacuation to control the number of evacuees on each path can be tracked by real-time positioning data. According to ASET = 7 min, $D_{\max} \in \{0.5, 1, 2, 3, 4, 5, 6\}$ corresponding evacuation models can accommodate more than 700 people. This is an increase of 300 people compared to no path capacity constraints and evacuation priority model. This paper draws the comparison of different settings based on Tables 6–8, as shown in Figures 13 and 14. $D_{\max} = +\infty$ people/m$^2$ means that there are no path capacity constraints.

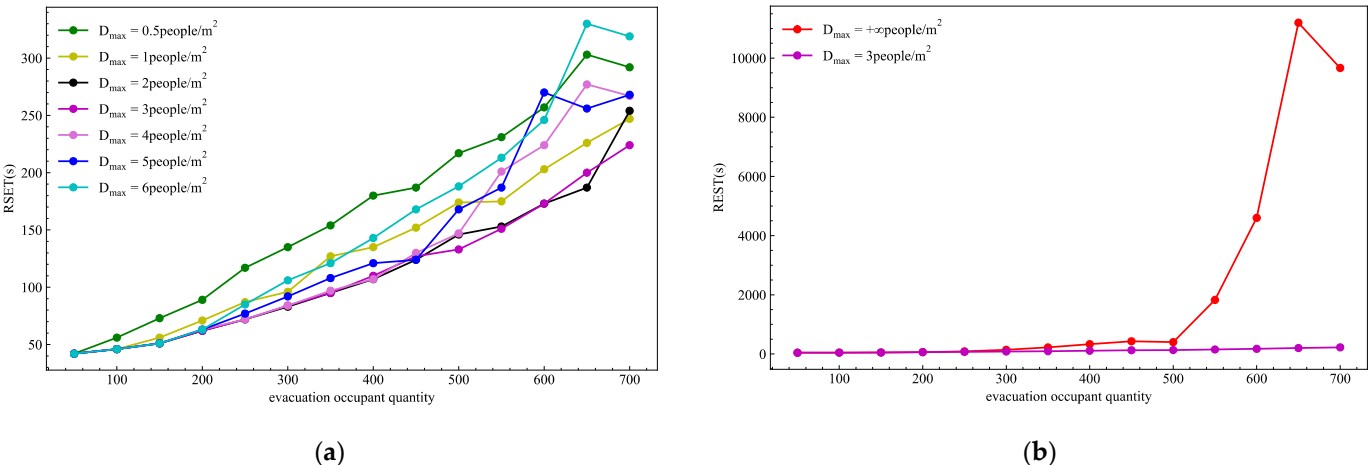

(**a**)                  (**b**)

**Figure 13.** The relationship corresponding to different $D_{\max}$ between RSET and the number of evacuees. (**a**) The polyline is about the influence of different $D_{\max}$ on RSET; (**b**) The polyline is about the influence of $D_{\max} = 3$ people/m$^2$ and no path capacity constraints on RSET.

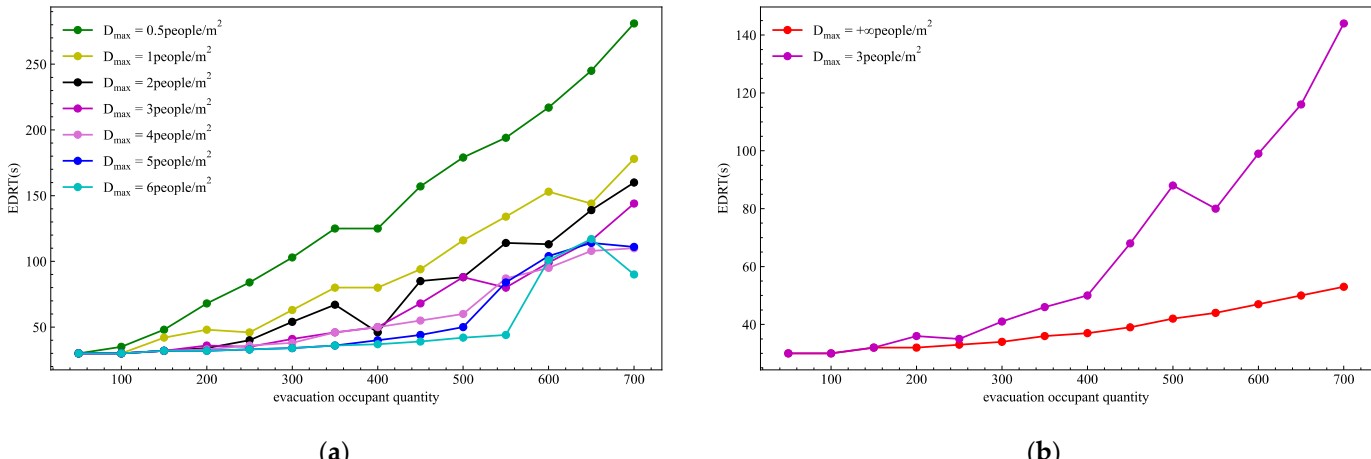

(**a**)                  (**b**)

**Figure 14.** The relationship corresponding to different $D_{\max}$ between EDRT and the number of evacuees. (**a**) The polyline is about the influence of different $D_{\max}$ on EDRT; (**b**) The polyline is about the influence of $D_{\max} = 3$ people/m$^2$ and no path capacity constraints on EDRT.

Figure 13a shows that $D_{max}$ can affect RSET according to the polyline trend, and different settings of $D_{max}$ have different effects on RSET. When the number of evacuees is 700, RSET with $D_{max} = 3\,\text{people}/\text{m}^2$ is 224 s, which is 196 s smaller than the set ASET = 7 min. In addition, it can be seen from Figure 14a that the overall growth rate of RSET is also the smallest, and the maximum capacity of the model within the safety limitation exceeds 700 people. Therefore, the authors of this paper believe that $D_{max} = 3\,\text{people}/\text{m}^2$ is the optimal value. Figure 13b shows that when the number of evacuees exceeds 500 people, the RSET corresponding to $D_{max} = +\infty\,\text{people}/\text{m}^2$ begins to rise significantly, and is much larger than the RSET corresponding to $D_{max} = 3\,\text{people}/\text{m}^2$. In the end, when the number of evacuees is 700, the corresponding RSET is 9666 s, which is far more than 224 s. This proves that the path capacity constraints can significantly reduce the RSET, the occurrence of congestion on the path, and provide higher safety for the evacuees.

However, it can be seen from Figure 14b that, when the number of evacuees exceeds 200, EDRT with $D_{max} = 3\,\text{people}/\text{m}^2$ is higher than $D_{max} = +\infty\,\text{people}/\text{m}^2$, and as the number of evacuees increases, the gap between the EDRT of the two models increases. When the number of people is 700, the EDRT with $D_{max} = 3\,\text{people}/\text{m}^2$ is 144 s, which is 91 s higher than EDRT with $D_{max} = +\infty\,\text{people}/\text{m}^2$. The above data can illustrate that, although the decision of path capacity constraints can reduce the evacuation time of all evacuees, it cannot guarantee the safety of evacuees on the dangerous path during the fire. Therefore, this paper adds evacuation priority on the basis of path capacity constraints.

### 4.3. Priority Evacuation Model with Path Capacity Constraints

Concerning when there are path capacity constraints and evacuation priority specified according to dynamic change data of the cruise ship risk level under fire situation in Table 3, this paper conducts a comparative analysis of the evacuation time for $D_{max} = \{0.5, 1, 2, 3, 4, 5, 6\}$. In this way, it is judged whether the evacuation time will be affected. The result of the evacuation is as shown in Tables 9 and 10.

**Table 9.** The relationship between the number of evacuees and RSET when $D_{max}$ is different.

| $N$ | 0.5 | 1 | 2 | 3 | 4 | 5 | 6 |
|-----|-----|-----|-----|-----|-----|-----|-----|
| 50 | 57 | 57 | 57 | 57 | 57 | 57 | 57 |
| 100 | 67 | 68 | 67 | 67 | 67 | 67 | 67 |
| 150 | 84 | 84 | 84 | 82 | 82 | 82 | 82 |
| 200 | 91 | 91 | 91 | 101 | 101 | 101 | 101 |
| 250 | 100 | 100 | 100 | 112 | 135 | 135 | 135 |
| 300 | 115 | 131 | 115 | 125 | 136 | 151 | 151 |
| 350 | 131 | 134 | 131 | 134 | 155 | 176 | 193 |
| 400 | 162 | 148 | 162 | 160 | 188 | 191 | 258 |
| 450 | 157 | 167 | 157 | 188 | 180 | 214 | 293 |
| 500 | 178 | 175 | 178 | 213 | 216 | 223 | 298 |
| 550 | 213 | 209 | 213 | 219 | 228 | 254 | 301 |
| 600 | 243 | 219 | 243 | 222 | 267 | 307 | 358 |
| 650 | 285 | 226 | 285 | 228 | 302 | 306 | 500 |
| 700 | 278 | 254 | 278 | 239 | 308 | 406 | 530 |

**Table 10.** The relationship between the number of evacuees and EDRT when $D_{\max}$ is different.

| N | 0.5 | 1 | 2 | 3 | 4 | 5 | 6 |
|---|---|---|---|---|---|---|---|
| 50 | 30 | 30 | 30 | 30 | 30 | 30 | 30 |
| 100 | 30 | 30 | 30 | 30 | 30 | 30 | 30 |
| 150 | 32 | 32 | 32 | 32 | 32 | 32 | 32 |
| 200 | 32 | 32 | 32 | 32 | 32 | 32 | 32 |
| 250 | 33 | 35 | 33 | 33 | 33 | 33 | 33 |
| 300 | 36 | 41 | 36 | 34 | 34 | 34 | 34 |
| 350 | 39 | 52 | 39 | 39 | 36 | 36 | 36 |
| 400 | 40 | 65 | 40 | 44 | 37 | 37 | 37 |
| 450 | 46 | 65 | 46 | 49 | 39 | 39 | 39 |
| 500 | 47 | 75 | 47 | 55 | 43 | 42 | 42 |
| 550 | 57 | 69 | 57 | 59 | 49 | 44 | 44 |
| 600 | 57 | 77 | 57 | 59 | 61 | 55 | 47 |
| 650 | 71 | 156 | 71 | 65 | 71 | 65 | 50 |
| 700 | 92 | 154 | 92 | 63 | 81 | 75 | 56 |

When $D_{\max} \in \{0.5, 1, 2, 3, 4\}$, the RSET in Table 9 increases within 80 s compared to the RSET in Table 7. When the number of evacuees is 650–700, the RSET with $D_{\max} \in \{5, 6\}$ in Table 9 increases by 200–250 s compared to the RSET with $D_{\max} \in \{5, 6\}$ in Table 7. After the evacuees have the different evacuation priority, the RSET will increase to different degrees based on different $D_{\max}$. The growth rate is not very obvious. This proves that the overall evacuation time has not been greatly affected after the evacuees' evacuation priority for $D_{\max} \in \{0.5, 1, 2, 3, 4\}$. In addition, when the number of evacuees is between 200 and 700, the EDRT in Table 10 is reduced within 200 s compared to the EDRT in Table 8. According to the data, the RSET and EDRT polyline corresponding to each $D_{\max}$ are drawn, as shown in Figures 15–17, comparing the RSET and EDRT of different evacuation models when $D_{\max} = 3 \, \text{people}/\text{m}^2$.

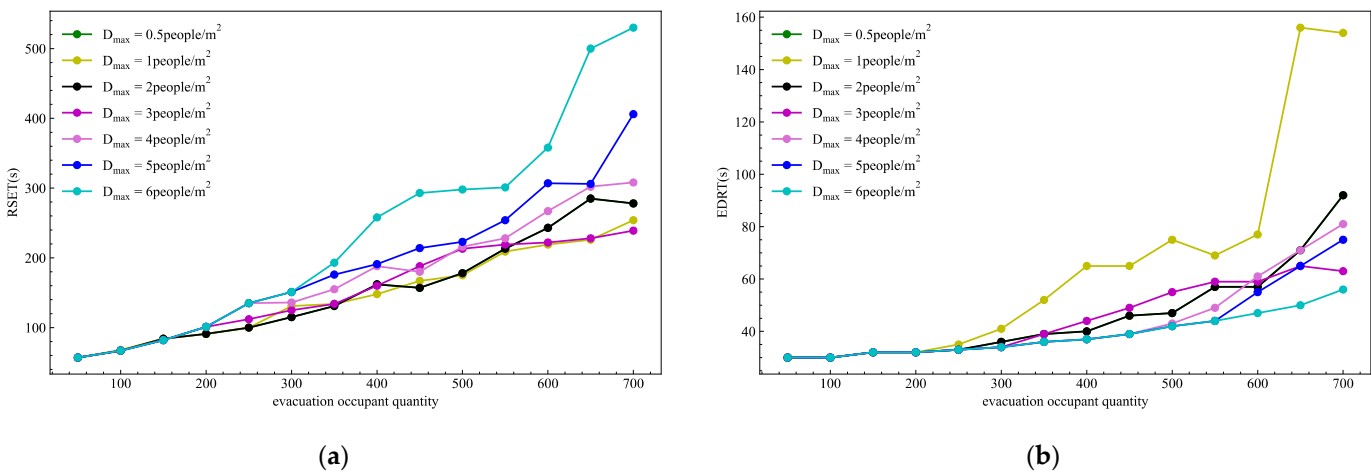

**Figure 15.** The influence of different $D_{\max}$ on evacuation time is set based on the evacuees' priority model. (**a**) The influence polyline of $D_{\max}$ on RSET; (**b**) The influence polyline of $D_{\max}$ on EDRT.

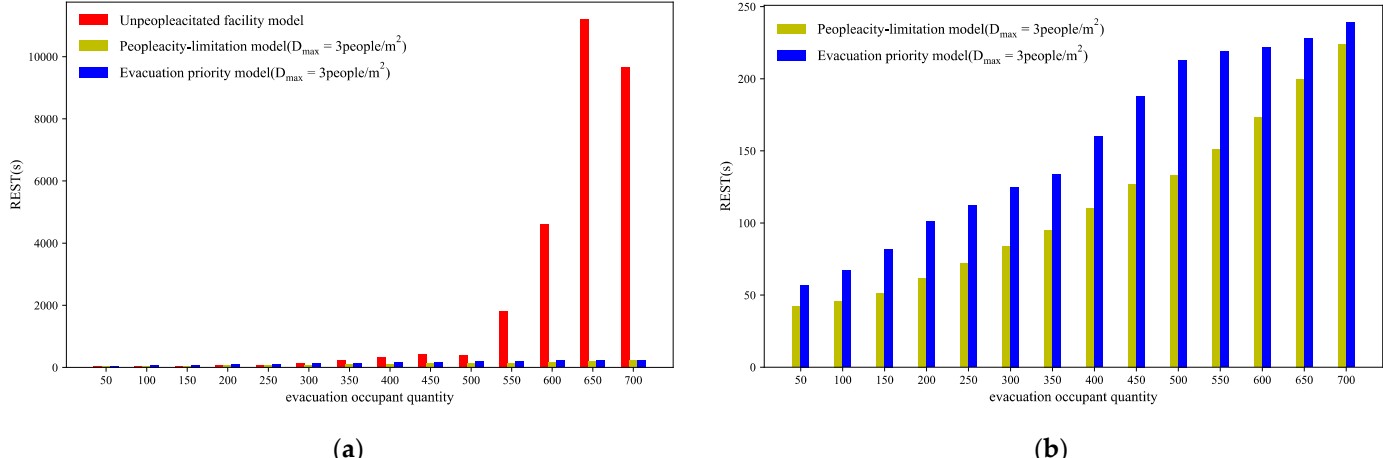

**Figure 16.** RSET comparison of evacuation simulations with different evacuation models. (**a**) RSET for evacuation simulation of three types of models; (**b**) Comparison of RSET between path capacity constraints and adding evacuees' priority based on the path capacity constraints.

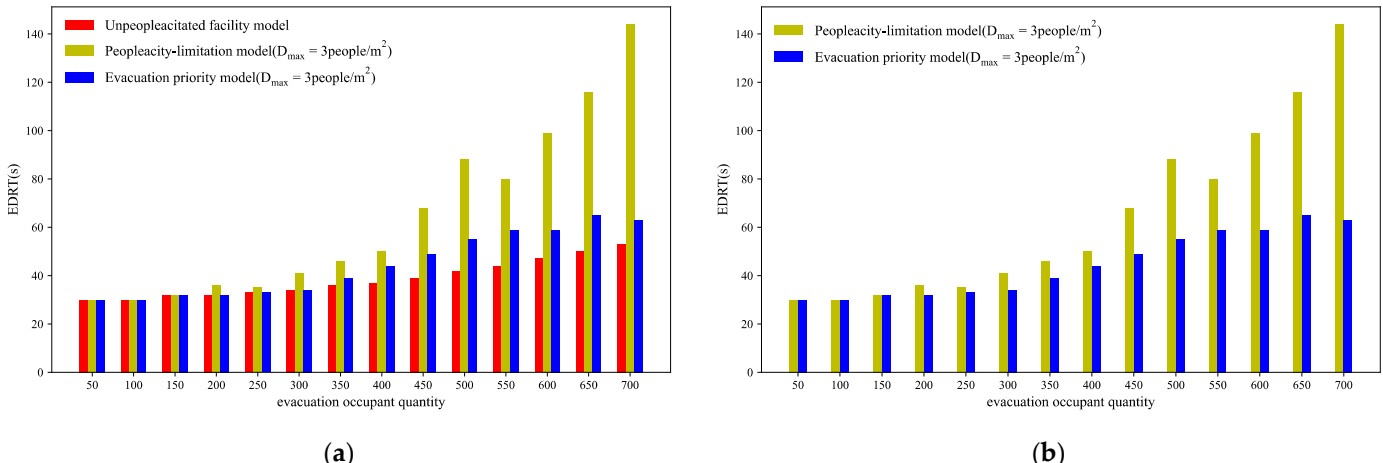

**Figure 17.** EDRT comparison of evacuation simulations with different evacuation models. (**a**) EDRT for evacuation simulation of three types of models; (**b**) Comparison of EDRT between path capacity constraints and adding evacuees' priority based on the path capacity constraints.

Figure 15 shows the impact of the model after adding the evacuees' priority on RSET and EDRT. As the number of evacuees increases, the RSET and EDRT tend to increase. In addition, RSET and EDRT are also related to $D_{max}$. When the number of evacuees exceeds 200, the RSET with $D_{max} = 6\,\text{people}/\text{m}^2$ will be longer than the RSET with $D_{max} \in \{0.5, 1, 2, 3, 4, 5\}$. When the number of evacuees exceeds 200, the EDRT with $D_{max} = 1\,\text{people}/\text{m}^2$ will be longer than the EDRT with $D_{max} \in \{0.5, 2, 3, 4, 5, 6\}$. When $D_{max} = 3\,\text{people}/\text{m}^2$, it will greatly save RSET and EDRT and improve system safety. In addition, it can be seen from Figure 16b that RSET will increase slightly. However, compared with the EDRT with only path capacity constraints from Figure 17b, EDRT with the priority evacuation model will be reduced and the corresponding RSET difference is not obvious. This proves that, when too many evacuees are evacuated, the priority evacuation model can ensure the safety of evacuees in high-risk paths and shorten their evacuation time while ensuring that the overall evacuation time is not greatly affected. The evacuation simulation effectively verifies the advantages of path capacity constraints and evacuees' priority evacuation. This is to ensure the rapid evacuation of evacuees from local dangerous

areas and the safety of all evacuees in the entire area, and to provide effective advice on evacuation strategies in actual evacuation.

## 5. Conclusions

Based on the fire emergency, this paper not only proposes a priority evacuation model with path capacity constraints, but also improves the program of evacuation network topology and evacuees' attribute changes. The fire simulation was carried out based on PyroSim. The evacuation simulation was carried out based on Python 3.6. Based on the obtained simulation data, it is found that, when a large number of evacuees are evacuated inside a cruise ship, the priority evacuation model with path capacity constraints reduces RSET to less than 600 s. The experimental results analyze that, when the number of evacuees exceeds 550, the evacuation time without path capacity constraints will exceed 1827 s. After the path capacity constraints are set, the maximum RSET with $N = 700$ is only 530 s. This proves that setting the path capacity constraints ensures the safety of overall evacuation. The difference $D_{max}$ in the maximum crowd density of the path will also affect RSET and EDRT. According to the experimental results, when $D_{max} = 3 \, \text{people}/\text{m}^2$, the RSET and EDRT are the shortest. In addition, when the number of evacuees is between 200 and 700, the safety of evacuees on dangerous paths will be further guaranteed. This model can not only improve evacuees' safety near the fire area, but can also prevent congestion in various paths.

In actual emergencies, the path capacity constraints and priority evacuation model proposed can provide guidance. When the number of evacuees is between 0 and 150, the path capacity constraints shall be set during evacuation to improve the evacuees' safety. When the number of evacuees is between 150 and 700, the priority evacuation model is adopted to ensure the safety of evacuees in local dangerous areas. In actual evacuation, measures such as smart bracelets, signs, rescuers, etc. can be used to realize the evacuation strategy in actual application about path capacity constraints and the priority evacuation.

**Author Contributions:** Conceptualization, Y.L. and H.Z.; methodology, Y.L.; software, Y.L.; validation, Y.L. and H.Z.; formal analysis, Y.Z.; investigation, K.D.; resources, L.D.; data curation, L.D.; writing—original draft preparation, Y.L.; writing—review and editing, Y.L.; visualization, Y.L.; supervision, H.Z.; project administration, H.Z.; funding acquisition, H.Z. All authors have read and agreed to the published version of the manuscript.

**Funding:** This research received no external funding.

**Institutional Review Board Statement:** Not applicable.

**Data Availability Statement:** In this research, data supporting reported results can be found at https://fseg.gre.ac.uk/validation/ship_evacuation (accessed on 11 October 2021).

**Conflicts of Interest:** The authors declare no conflict of interest.

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
