# Peer review of "Evacuation Strategy Considering Path Capacity and Risk Level for Cruise Ship"

_jmse, doi:10.3390/jmse10030398_

Round 1

Reviewer 1 Report

Comments

The evacuation model is promising, and the aim of the work is well described, and many theoretical gaps have been improved. The paper needs improvements in the English sentence and grammatical construction - some changes are listed in the comments below. The explanation of the different algorithms that constitute the basis of the methodology can be improved. It will be good if a native English editor proofreads the paper, especially the Abstract. Thank you.

Lines 8-14 – Sentence construction for abstract is poor – please rewrite. Abstract is most important to enable the reader to glean the aims, significance, and implication of the research.

Line 91 – ‘In order to validate the evacuation model…’ is more appropriate

Line 237 – ‘After literature research [39-42], the research…’ the term research is repeated, preferable to use ‘literature review’ in this case

Line 378 – The sentences ‘In order to verify….’ and ‘This paper completes …’ must be linked

Line 537 – The sentence beginning with ‘The improvement of the program…’ needs to be rewritten.

Author Response

Authors Response to the Review of jmse-1583063

Dear reviewer,

Thanks very much for your careful reading and insightful comments. We are pleased to answer the suggestions and the manuscript (Manuscript No.: jmse-1583063) has also been extensively revised according to the comments, and the detailed corrections are listed below:

Reviewer: 1

Comment 1: Lines 8-14 – Sentence construction for abstract is poor – please rewrite. Abstract is most important to enable the reader to glean the aims, significance, and implication of the research.

Reply: Thank the reviewer for the suggestions. This article rewrites the abstract on line8-23.

Comment 2: Line 91 – ‘In order to validate the evacuation model…’ is more appropriate

Line 237 – ‘After literature research [39-42], the research…’ the term research is repeated, preferable to use ‘literature review’ in this case

Line 378 – The sentences ‘In order to verify….’ and ‘This paper completes …’ must be linked

Line 537 – The sentence beginning with ‘The improvement of the program…’ needs to be rewritten.

Reply: Thanks for the reviewer's suggestion. We have corrected these four grammatical errors on line94, line243, line375 and line533.

(1) In order to verify the evacuation model in this paper and analyze the advantages and disadvantages of the model performance, it is necessary to investigate and obtain evacuation scene data and build an evacuation network topology for the evacuation scene.

(2) After literature research [34-38], the literature review on the relationship between evacuees’ moving speed and crowd density is summarized and analyzed, as shown in Figure 4, Figure 5, and Figure 6.

(3) In order to verify the effect of priority evacuation model on the emergency evacuation of cruise ships, as well as the impact of the path capacity constraints on the evacuation time, this paper completes three types of evacuation models for evacuation simulation based on Python.

(4) Based on the fire emergency, this paper not only proposes a priority evacuation model with path capacity constraints, but also improves the program of evacuation network topology and evacuees’ attributes changes.

Author Response

Authors Response to the Review of jmse-1583063

Dear reviewer,

Thanks very much for your careful reading and insightful comments. We are pleased to answer the suggestions and the manuscript (Manuscript No.: jmse-1583063) has also been extensively revised according to the comments, and the detailed corrections are listed below:

Reviewer: 2

Comment 1: Path capacity and risk level factors are the main factors in this contribution. However, the article does not include any previous literature or contributions relating to these factors. In addition, the authors did not mention whether or not other researchers applied the priority strategy to the evacuation process in previous studies.

Reply: Thank the reviewer for the suggestions. Indeed, we did not introduce studies on link path capacity and risk level in the Introduction. This is because researchers usually consider the path capacity and risk level when studying route planning, and the starting point of this paper is an idea of controlling the flow of evacuees. This is similar to the control of traffic flow. Therefore, we include the research background on the path capacity and risk level in the third natural paragraph of the introduction. We rewrote the introduction on line54-68.

Comment 2: Organization of the paper in Sec 1 is missing.

Reply: Thank the reviewer for the suggestions. We add to organization of the paper on line87-92.

Comment 3: Is the work that includes equations (3, 4, 5 and 6) knowledge found in reference [34]? If so, keep it in an appendix. If you have a contribution to this work, show it.

Reply: Thank the reviewer for the suggestions. These formulas 3, 4, and 5 come from reference [29]. Formula 6 is proposed by us to divide the risk level of paths. We have added citations in the text on line172.

Comment 4: I did not find the reference [34].

Reply: Thank the reviewer for the suggestions. This paper is a Chinese master's paper, the download link is https://oss.wanfangdata.com.cn/file/download/degree_Y1719612.aspx.

https://d.wanfangdata.com.cn/thesis/ChJUaGVzaXNOZXdTMjAyMTEyMDESCFkxNzE5NjEyGgh0Yms3djZ2cg%3D%3D.

Comment 5 and 6: Regarding the equation 6, What is the value of the danger factor when vis< 0.6 but < 0.25? Regarding the equation. 6, there is a union operation between predicates. Shouldn't it be "OR"?

Reply: Thank the reviewer for the suggestions. There is a real problem with the formulation of Equation 6 here. What we want to express is that the risk level of the path is determined by the value corresponding to the highest risk index among the three indicators. There is no problem with the following fire simulation data, we have made changes to equation 6 on line186.

 (6)

Comment 7: Prove that the integration of speed-density models by calculating the average movement speed sets for scholar [39-41] results in an improved relationship of speed-density to emergency situations. Justify scientifically why your approach (taking the average) is a solution to the large difference in the speed of the evacuees? Could the speed be higher or lower than what was produced from all the references mentioned? Keep in mind that some of these forms are also not for emergency evacuation.

Reply: Thank the reviewer for the suggestions. The density-velocity relations obtained by these researchers based on experimental data are all based on different evacuation scenarios, so there are differences in the obtained density-velocity relations. Especially for the larger density and the smaller density corresponding velocity numerical difference is obvious. We combined their experimental data by taking an average to make the density-velocity relationship closer to the actual evacuation situation. For example, the evacuation velocity should not be 0 when the density is high, and there should be an upper limit when the density is low. Fitting velocity models by different scholars is a new method and we give a more detailed explanation on page 9 (lines 255-264).

Comment 8: The method of fitting the speed-density model of evacuees has been found in undergraduate courses, so this method need not be presented. Just give equation 16. Similarly, Dijkstra algorithm is available in undergraduate courses. I think no need to explain the method from line 354 to 377.

Reply: Thank the reviewer for the suggestions. This paper retains the content of the curve fitting part, because it covers some new data and methods, and cuts the content of the Dijkstra algorithm on line361.

Comment 9: In figure 10, the pseudocode displays that if priority =1 and length(x,nn) >= 2, then order(x) =1. However, if in this case there are people in higher priority going to the next path, the flowchart displays that order(x)=0. Therefore, pseudocode and flowchart do not match each other.

Reply: Thank the reviewer for the suggestions. There is indeed a logical error in the flowchart here, and we have corrected it in the corresponding position of the flowchart on line 353.

Comment 10: Why you choose value 2m in the if statement? What does it mean?

Reply: Thank the reviewer for the suggestions. When the priority evacuation is performed considering the risk level of the path, evacuees on the path with lower risk level needs to wait for the evacuees on the path with higher risk level to evacuate first. To ensure that evacuees on lower risk paths can continue to move away from the intersection, a value of 2m was chosen as the threshold for queuing. When the evacuees are far away from the node they are currently heading to, they keep moving forward. we add a detailed explanation on page 13 (lines 358-360)

Comment 11: What is the  and in the pseudocode in Figure 11? What does  means?

Reply: Thank the reviewer for the suggestions. We got the letters wrong here by our negligence.

 and  correspond to  and  in Table 1, respectively.  means the shortest route to the exit of node i and  means the shortest route length to the exit of node i.  also has the same error.  means the evacuation path of the evacuee adopts the shortest route of the first node. We have made corrections to the pseudocode and flowchart on Page 15(line383).

Comment 12: In the pseudocode in figure 11:

Does this comparison apply to every repetition of the movement? Or only when initializing the decision to follow the shortest path? In other words, suppose x goes from node i to j, according to dijkstra, the node i is further away from the exit than node j. node j is the end node of x(ij).

Reply: Thank the reviewer for the suggestions. this comparison applies to every repetition of the movement. The shortest distance from each node to the exit will be calculated and stored offline. The function of this step is to determine which direction the evacuee goes to the path has the shortest total route length. x.i and x.j are explained in Table 4, they represent the first and end nodes of the path where the evacuee x is located.

Comment 13: Provide an overall flow chart.

Reply: Thank the reviewer for the suggestions. We added a complete flowchart on line110.

Round 2

Reviewer 2 Report

Authors should discuss relevant contributions in some literature that has considered the underlying factors addressed in the article. See for examples contributions such as in:

  • https://doi.org/10.1016/j.jobe.2020.101687
  • https://www.researchgate.net/publication/2561579
  • Dynamic Evacuation Planning on Cruise Ships Based on an Improved Ant Colony System (IACS). Journal of Marine Science and Engineering 2021.

Equation 6 is incorrect. It's better to write it as pseudo code if the authors can't write it correctly as an equation.

A total flow chart is recommended which includes the flow in Fig. 11 and 12. The flow chart in figure 2 is not significant.

Choosing a value of 2 m as a distance is not justified. Please state if it is assumption or there is literature suggested this value. What is the effect of the value if it is more or less?

The logic of pseudocode in Fig. 11 and 12 is not correct.

The exponential function for speed-density relation contributed by the authors is already in ref.:

Zhang, X. Study on rapid evacuation in high-rise buildings. Eng. Sci. Technol. 2017, 20, 1203–1210. [CrossRef]

Author Response

Authors Response to the Review of jmse-1583063

Dear reviewer,

Thanks very much for your careful reading and insightful comments. We are pleased to answer the suggestions and the manuscript (Manuscript No.: jmse-1583063) has also been extensively revised according to the comments, and the detailed corrections are listed below:

Reviewer: 2

Comment 1: Authors should discuss relevant contributions in some literature that has considered the underlying factors addressed in the article. See for examples contributions such as in:

  • https://doi.org/10.1016/j.jobe.2020.101687
  • https://www.researchgate.net/publication/2561579
  • Dynamic Evacuation Planning on Cruise Ships Based on an Improved Ant Colony System (IACS). Journal of Marine Science and Engineering 2021.

Reply: Thank the reviewer for the suggestions. We refer to these three literatures, who developed corresponding heuristics for fire conditions and path capacity. Their algorithm can get the best evacuation route, and we can further optimize the entire evacuation process by referring to their evacuation environment and combining the personnel behavior planning in this paper. We have rewritten the introduction to address these issues on line23-67.

Comment 2: Equation 6 is incorrect. It's better to write it as pseudo code if the authors can't write it correctly as an equation.

Reply: Thank the reviewer for the suggestions. We converted Equation 6 into the form of pseudocode on line179-181.

Comment 3: A total flow chart is recommended which includes the flow in Fig. 11 and 12. The flow chart in figure 2 is not significant.

Reply: Thank the reviewer for the suggestions. We deleted the flow chart of Figure 2 and integrated Figures 11 and 12 together on line324-325.

Comment 4: Choosing a value of 2 m as a distance is not justified. Please state if it is assumption or there is literature suggested this value. What is the effect of the value if it is more or less?

Reply: Thank the reviewer for the suggestions. We modified the flowchart in Figure 11 and replaced 2m with .  represents the threshold for evacuees to stay and wait. Only when the distance from the evacuees to the next node is less than this value, the evacuation priorities need to be compared between evacuees, otherwise they keep moving forward. This paper takes 2m as a hypothesis to complete the subsequent evacuation simulation experiments.

Comment 5: The logic of pseudocode in Fig. 11 and 12 is not correct.

Reply: Thank the reviewer for the suggestions. We have removed the pseudocode for Figure 11 and Figure 12.

Comment 6: The exponential function for speed-density relation contributed by the authors is already in ref.:

Zhang, X. Study on rapid evacuation in high-rise buildings. Eng. Sci. Technol. 2017, 20, 1203–1210. [CrossRef]

Reply: Thank the reviewer for the suggestions. Zhang, X. proposed and studied a new evacuation device for high rising buildings in fire accident. This device mainly consisted of special spiral slideway and shunt valve. People in this device could fast slide down to the first floor under gravity without any electric power and physical strength, which is suitable for various emergency evacuation including mobility impaired persons.

High crowd density means slow moving speed, we proposed an exponential function including straight path and stairs speed characteristics to illustrate the relationship between crowd density and moving speed. The data used in this method comes from the experiments of various scholars, which is not the same as the evacuation device. In addition, the curve fitted by the exponential function in this paper can be better combined with the actual situation than the curve fitted by other methods.

Round 3

Reviewer 2 Report

The paper is in acceptable condition. If the authors return Figure 11 to its original state (Figure 11 and 12) it will be even better

This manuscript is a resubmission of an earlier submission. The following is a list of the peer review reports and author responses from that submission.

Round 1

Reviewer 1 Report

The solution of the optimal path finding in fire evacuation problems should be in a direct connection with an opportunity to use a decision in real life. Here such connection is absent:

- paths a priory (before evacuation simulation) are assigned with some estimates of danger without simulation the spread of the fire;

- who does know the ‘’weight” of each path in real evacuation scenario to take this path or not to take to realize the optimal one from simulation point of view?

- Dmax is an input parameter of the model, but at the same time is not controllable under evacuation conditions and self-adaptive;

- there is no opportunity to reproduce computational are as it, but shape of the ways and doors on evacuation path  

So how to apply the model to real task if it “plays” under artificial rules which strongly determines a result?

How to use results obtained?

More over ASET is not unique value for the building. It depends on place of the fire, design of the building, fire load (other conditions) and along the evacuation paths in different control points ASET and RSET should be estimated and compared. And one should do such comparisons for number of places of fire in the building.

So I see the article as a simulation exercise which need to be improved as a simulation tool without true real application value.

Author Response

Authors Response to the Review of jmse-1467077

Dear reviewer,

Thanks very much for your careful reading and insightful comments. We are pleased to answer the suggestions and the manuscript (Manuscript No.: jmse-1467077) has also been extensively revised according to the comments, and the detailed corrections are listed below:

Reviewer: 1

Comment 1: The solution of the optimal path finding in fire evacuation problems should be in a direct connection with an opportunity to use a decision in real life. Here such connection is absent:

Reply: Thank the reviewer for the suggestions. This paper focuses on explaining the impact of the path capacity constraints and priority on evacuation. This paper gives a clear explanation for the method of combining the path capacity constraints and priority model with the actual situation on page 18. (line462-465).

For the actual evacuation situation, the method of capacity constraints can be used for reference, and the evacuees can be equipped with smart bracelets, signs and crews who direct the evacuation to control the number of evacuees on each path.

Comment 2: paths a priory (before evacuation simulation) are assigned with some estimates of danger without simulation the spread of the fire:

Reply: Thanks for the reviewer's suggestion. In practice, the risk level of each path should be judged by the CO concentration detector, temperature detector and smoke visibility detector of each path. This paper mainly analyzes and verifies the experimental results of the path capacity constraint and evacuation priority model. The evacuation model can also be dynamically changed, so our numerical analysis of fire spread will be studied in the future. We give an explanation on page 22 (line561-563).

In addition, this article does not consider the dynamic spread of the fire and the dynamic change of the corresponding path dangerous degree, which will be carried out in the follow-up research.

Comment 3: who does know the ‘’weight” of each path in real evacuation scenario to take this path or not to take to realize the optimal one from simulation point of view?

Reply: Thank the reviewer for the suggestions. The weight of each path in the actual evacuation scene can be calculated in advance and the actual evacuation can be guided by smart bracelets, signs and crews. Through the positioning device on the cruise ship, the number of people on different paths can be monitored in real time. This data is then passed to the crew, and the crew combines these data to allocate evacuation routes. In the revised manuscript, we give a clearer expression of how the path width is defined in the common situation with different widths along a path on page 3 (line134-137).

represents the length of the path, and its value is the distance between the first node and end node. represents the width of the path, and its value is the shortest distance both sides of the path.

Comment 4: Dmax is an input parameter of the model, but at the same time is not controllable under evacuation conditions and self-adaptive.

Reply: Thank the reviewer for the suggestions. The data is provided to the crew by the way of positioning the bracelet, and then the crew will command. To control the maximum capacity Dmax in the path. This paper gives a clear explanation for path capacity constraints on page 10. (line295-300).

The capacity constraints of the path can be converted into the maximum crowd density setting of the path, see formula (13).

                             (13)

represents the maximum evacuees’ capacity on the path. represents the maximum crowd density of the path. By formula (13), the constraints on total number of evacuees on all paths can be converted into an equal maximum crowd density .

Comment 5: there is no opportunity to reproduce computational are as it, but shape of the ways and doors on evacuation path.

Reply: Thank the reviewer for the suggestions. We have all the procedures here, and We can provide them all if needed.

Comment 6: So how to apply the model to real task if it “plays” under artificial rules which strongly determines a result? How to use results obtained?

Reply: Thank the reviewer for the suggestions. The fire situation is detected by sensors, and then a reasonable evacuation route is provided for different people based on the flow of people on different paths. When the number of evacuees is large, priority evacuation and the path capacity constraints model can be used to evacuation, and the maximum path capacity constraints is selected as 2people/m². When the number of evacuees is small, just follow the shortest route to evacuate. When number of evacuees is large, the evacuation decision is limited by the capacity of the path. The evacuation of different the number of evacuees and fire simulations in different locations can be used as a backup plan, and finally the crew is used to direct the evacuation. After the Titanic crew led the passengers, they finally left. Simulate multiple schemes offline first, and then select the corresponding scheme during emergency evacuation. We have rewritten the conclusions and experimental analysis in the paper on page 22(line554-560).

In actual emergencies, the path capacity constraints and priority evacuation model proposed in this paper can provide guidance. When the number of evacuees is between 0 and 600, the path capacity constraints shall be set during evacuation to improve the safety of evacuees. When the number of evacuees is between 650 and 850, the priority evacuation model is adopted to ensure the safety of evacuees in local dangerous areas. In actual evacuation, measures such as smart bracelets, signs, rescuers, etc. can be used to provide path capacity constraints and the priority evacuation.

Comment 7: More over ASET is not unique value for the building. It depends on place of the fire, design of the building, fire load (other conditions) and along the evacuation paths in different control points ASET and RSET should be estimated and compared. And one should do such comparisons for number of places of fire in the building.

Reply: Thank the reviewer for the suggestions. In the revised manuscript, we give a more detailed explanation on page 16. (line409-428).

For fires in buildings, the available safe egress time (ASET) represents the time required for the building to reach the durability constraints. The fire resistance rating is a grading scale that measures the fire resistance of a building. It is determined by the combustion performance and fire resistance limit of the components that make up the building. The fire resistance rating of a building is one of the most basic measures in the fire protection technical measures specified in the building design fire protection code. The factors that affect the selection of fire resistance levels are: the importance of the building, the nature of use and fire hazard, the height and area of the building, the size of the fire load and other factors. The main building components of Class I fire-resistant buildings are all non-combustible. The main building components of Class II fire-resistant buildings are non-combustible except for the ceiling which is difficult to burn. The roof load-bearing components of Class III fire-resistance buildings are flammable. The fire wall of the Class â…£ fire-resistance building is non-combustible, and the rest are flame-retardant and flammable. For civil buildings and high-rise buildings with Class I and II fire resistance, ASET generally takes 5-7 minutes; for Class III and IV fire resistance ASET buildings, 2-4 minutes. The materials in cruise ships are generally non-flammable, so this article uses ASET=7minutes as a reference indicator to determine the maximum personnel capacity within the safety limits of different evacuation models. [46]. The relationship between the total number of evacuees and the evacuation time drawn according to the data in Table 4 is shown in Figure 12

Comment 8: So I see the article as a simulation exercise which need to be improved as a simulation tool without true real application value.

Reply: Thank the reviewer for the suggestions. Fire simulation is more complicated. This article mainly intends to verify the feasibility of path capacity constraints and priority evacuation through experiments. The fire numerical analysis and the dynamic change of the model are the next steps. In the revised manuscript, we give an explanation on page 22 (line561-563).

In addition, this article does not consider the dynamic spread of the fire and the dynamic change of the corresponding path dangerous degree, which will be carried out in the follow-up research.

Reviewer 2 Report

The aim of the paper is clearly stated and the supporting literature review is described. There are oversimplifications in the description of some parameters (e.g. RSET, which is composed of different elements that are not listed in the paper) and in some choices (e.g.: the choice of 10 m for the definition of the dangerous areas is not supported) and the methodology is not well stated. Some passages are too simplistic in the context of fire risk evaluation. Several sentences need revision as they are not clear. Grammar and proper language rules should be applied to the whole paper, in addition to properly expanding different abbreviations. The generalization of the results to be applied to other building types needs to be demonstrated. The novelty of the paper and how it fills a gap in literature need to be highlighted. The conclusions and results need to be re-organized and re-written as the language used is vague and not quantified. Use numbers instead of using generic terms like “slightly”, “too large”, “optimal” etc. These terms do not indicate accurate discussion of the results and the conclusion. Please refer to specific comments below.

Specific comments

Line 15: Please explain the acronym ‘IFC’

Line 46: BIM “Building Information Modelling” and IFC “Industry Foundation Classes” abbreviations need to be expanded

Line 47: GNG stands for geometric network graph and not geometric network diagram

Line 85, 96:  Please clarify specific literature from the SAFEGUARD Literature is used for this study

Line 123:  Please explain what ‘prone to fire’ means. A fire scenario identification and selection process would be required.

Line 133:  Please explain how the path width is defined in the common situation where different widths can be given along a path

Table 1: Please define Safe, Danger, Exit and Fire node

Table 2:  How is the risk level assessed?

Table 3: Please provide units of measurement for length and width

Lines 159-161: The applicability of the cruise ship model to other large buildings needs to be explained and verified

Line 180: Please provide explanation for the sentence ‘The ASET has more safety margin than the RSET’

Line 197:  Please note that the sentence beginning with ‘Because the speed…’ is missing a part.

Line 199:  Please explain why making the average of the different theoretical model provides results that are ‘closer to the actual situation’. Here a discussion on the validation of the proposed model could also be added, is the method taken from literature or is it new? In this last case it should be validated against real evacuation data

Line 224:  Please explain how the values in the equations are obtained

Line 262: Please provide an alternative representation of the algorithm used, the pseudocode is difficult to read, maybe a flow chart.

Line 302: Please note that the sentence beginning with ‘Because the speed…’ is missing a part.

Line 309:  Please explain why a distance of 10 meters is chosen for identifying danger nodes. There appears to be no relationship between this value and the previously cited risk assessment technique from Zhu et al.

Line 380: Python: please provide references

Line 384:  Please provide an alternative representation of the algorithm used, the pseudocode is difficult to read, maybe a flow chart.

Line 407: Clarify why the number of evacuees of 600 is chosen analytically. It is mentioned that the model will cause a surge at this value, but how this surge is determined is not clear

Line 411: Please explain what is meant with class I, II, III and IV fire resistance and add some sources for the ASET values reported

Line 442:  Please explain how the number of evacuees can be controlled in actual evacuations

Line 476:  Please explain what is meant with path equipped with capacity constraints and evacuation priority

Line 500-524: Use numbers instead of using generic terms like “slightly”, “too large”, “optimal” etc. These terms are vague and do not indicate accurate discussion of the results and the conclusion.

Line 522:  REST should be RSET

Author Response

Authors Response to the Review of jmse-1467077

Dear reviewer,

Thanks very much for your careful reading and insightful comments. We are pleased to answer the suggestions and the manuscript (Manuscript No.: jmse-1467077) has also been extensively revised according to the comments, and the detailed corrections are listed below:

Reviewer: 2

Comment 1: Please explain the acronym ‘IFC’ and BIM “Building Information Modelling”. IFC “Industry Foundation Classes” abbreviations need to be expanded and GNG stands for geometric network graph and not geometric network diagram.

Reply: Thank the reviewer for the suggestions. In the revised manuscript, we have corrected on page 2 (line 46 and line 48).

Building Information Modeling (BIM) to Industry Foundation Classes (IFC)

geometric network graph (GNG).

Comment 2: Please clarify specific literature from the SAFEGUARD Literature is used for this study.

Reply: Thank the reviewer for the comments and the suggestions. In the revised manuscript, we use https://fseg.gre.ac.uk/validation/ship_evacuation as a reference for accessing cruise ship data on page 2. (line86 and line97).

This paper completes the evacuation simulation on the cruise data set given in the European "SAFEGUARD" project [33].

[33] FIRE SAFETY ENGINEERING GROUP. Available online: https://fseg.gre.ac.uk/validation/ship_evacuation (accessed on 11 October 2021).

Comment 3: Please explain what ‘prone to fire’ means. A fire scenario identification and selection process would be required.

Reply: Thank the reviewer for the suggestions. In the revised manuscript, we give a clearer expression of the fire scenario identification and selection process on page 3 (line122-128).

In addition, in order to study the impact of path capacity constraints and evacuation priority on evacuation, it is necessary to set up fire locations and escape exits to further meet the requirements of evacuation simulation. In this project, the personnel designated the escape area as assembly area C. This paper uses the node leading to assembly area C as an exit. The materials in the ship are generally non-flammable, and the place where a fire occurs is often the luggage of tourists. Therefore, this paper uses the storage area as the place where the fire broke out.

Comment 4: Please explain how the path width is defined in the common situation where different widths can be given along a path.

Reply: Thank the reviewer for the suggestions. In the revised manuscript, we give a clearer expression of how the path width is defined in the common situation with different widths along a path on page 3 (line134-137).

represents the length of the path, and its value is the distance between the first node and end node. represents the width of the path, and its value is the shortest distance both sides of the path.

Comment 5: Table 1: Please define Safe, Danger, Exit and Fire node.

Reply: Thank the reviewer for the suggestions. In the revised manuscript, we moved the content of the dangerous node in the third section to the bottom of the Table 1 on page 5 (line164-172).

The steps for judging the risk level of a path are as follows:

Calculate all the distances from node i to fire node, see formula (3).

                   (3)

Judge whether the distance between the node and the fire node is less than 10 meters, or whether it is connected to the fire node. If any one of the above conditions is met, the type of the node is assumed to be a danger node. This type of node is marked in orange in Figure 2. The definition is shown in formula (4).

                 (4)

After setting up the fire node, exit node and danger node, the remaining nodes are all safe nodes.

Comment 6: Table 2: How is the risk level assessed?

Reply: Thank the reviewer for the suggestions. In the revised manuscript, we moved the content of the dangerous node in the third section to the bottom of the Table 1 on page 5 (line172-175).

Define the paths formed by connecting the fire node and the danger node or by the danger node and the danger node, has . Paths formed by connecting danger node and safe node, has . Paths formed by connecting safe node and safe node, has .

Comment 7: Table 3: Please provide units of measurement for length and width.

Reply: Thank the reviewer for the suggestions. Due to our negligence, we forgot to add a unit. We added the unit in Table 3 on page 6 (line194).

Path number

Length and width(m)

path number

Length and width(m)

Comment 8: The applicability of the cruise ship model to other large buildings needs to be explained and verified.

Reply: Thank the reviewer for the suggestions. In the revised manuscript, we give clear explanations and the literature of on the application of this model to other large buildings on page 5 (line176-180).

When many large buildings are used as evacuation scenarios to construct evacuation network topology, many attributes of nodes and paths can also refer to the extended attributes of this paper, such as crowd density, shortest route, node type, and path risk level. These extended attributes are conducive to the research of evacuation algorithms and can be applied to large buildings with similar evacuation structures[35].

Comment 9: Please provide explanation for the sentence ‘The ASET has more safety margin than the RSET’

Reply: Thank the reviewer for the suggestions. In the revised manuscript, we give a clearer expression of the ASET has more safety margin than the RSET on page 6 (line199-204).

As the fire situation continues to develop, the time during which the building reaches the endurance limit is called the available evacuation time. ASET includes evacuation start time, evacuation movement time, and safety margin. RSET only includes the evacuation start time and the evacuation movement time Therefore, The ASET has more safety margin than the RSET.

Comment 10: Please note that the sentence beginning with ‘Because the speed…’ is missing a part.

Reply: Thank the reviewer for the suggestions. In the revised manuscript, we added a note before the introduction section of speed on page 6 (line208-211).

After completing the construction of the evacuation network topology, it is also necessary to obtain the moving speed of the personnel when evacuating in different situations. When the evacuation speed is closer to the actual evacuation speed, the evacuation time is more accurate.

Comment 11: Please explain why making the average of the different theoretical model provides results that are ‘closer to the actual situation’. Here a discussion on the validation of the proposed model could also be added, is the method taken from literature or is it new? In this last case it should be validated against real evacuation data.

Reply: Thank the reviewer for the suggestions. Fitting speed models of different scholars is a new method, we give a more detailed explanation on page 7-8 (line224-236).

Figures 3, 4, and 5 respectively show the relationship between the moving speed of evacuees on different paths. Since the experimental formulas obtained by different scholars are based on experimental data obtained in different scenarios, this situation has led to a large difference in their speed models within a certain interval of crowd density. It can be seen from Figure 3 that when the crowd density exceeds 4people/m², Fruin and Nelson and Mowrer models will no longer be applicable. In the actual situation, when the crowd density is too high, the people moving speed will be very small but exist. Therefore, their speed model is not perfect. When the population density is close to 0 people/m², there are big differences between scholars’ models. In order to reduce the impact of the differences between these models and perfect the personnel speed formula, this paper will synthesize the speed models of various scholars to reduce the impact of differences caused by different scenarios and make the application of the formula wider to keep it close to the actual situation.

Comment 12: Please explain how the values in the equations are obtained.

Reply: Thank the reviewer for the suggestions. In the revised manuscript, we have added solving formulas (10) and (11) on page 6 (line259-262).

This paper uses the least square method to solve the optimal solution of the linear programming model, as shown in equations (10), (11).

                     (10)

           (11)

Comment 13: Please provide an alternative representation of the algorithm used, the pseudocode is difficult to read, maybe a flow chart.

Reply: Thank the reviewer for the suggestions. In the revised manuscript, we have added a flow chart on page 11 (line318).

Comment 14: Please note that the sentence beginning with ‘Because the speed…’ is missing a part.

Reply: Thank the reviewer for the suggestions. In the revised manuscript, this paper focuses on verifying the impact of path capacity constraints and priority on the evacuation results, so this paper deletes this part.

Comment 15: Please explain why a distance of 10 meters is chosen for identifying danger nodes. There appears to be no relationship between this value and the previously cited risk assessment technique from Zhu et al.

Reply: Thank the reviewer for the suggestions. In the revised manuscript, this paper focuses on verifying the impact of path capacity constraints and priority on the evacuation results, so this paper deletes this part. In addition, this paper gives an explanation of 10m on page 5(line158-163)

This paper focuses on the impact of setting path capacity constraints and evacuation priorities on the evacuation results. Therefore, this paper uses the distance between all areas and the fire area to judge the risk level of all paths, thereby simplifying the operation. After investigation[34], when the smoke exhaust system is in good working condition, the smoke visibility is generally 10m. In this paper, the distance of 10m is used as a reference index.

Comment 16: Python: please provide references.

Reply: Thank the reviewer for the suggestions. In the revised manuscript, we use https://www.python.org/getit as a reference for software version on page 14. (line389 and line390).

The software version used in the experiment is Python 3.6[45] and Pycharm 2019.3.3x64.

[45] Python. Available online: https://www.python.org/getit (accessed on 15 June 2020).

Comment 17: Please provide an alternative representation of the algorithm used, the pseudocode is difficult to read, maybe a flow chart.

Reply: Thank the reviewer for the suggestions. In the revised manuscript, we added a flow chart on page 15. (line393).

Comment 18: Clarify why the number of evacuees of 600 is chosen analytically. It is mentioned that the model will cause a surge at this value, but how this surge is determined is not clear.

Reply: Thank the reviewer for the suggestions. When the number of evacuees is 600, the evacuation time will increase sharply. This sentence is judged by the slope of the polyline in the Figure12. However, after careful inspection, it is found that it is more appropriate to replace 600 with 500. We gave a detailed description on page 16. (line431-443).

It can be analyzed from Figure 12 that the rate of increase in RSET is not obvious before the number of evacuees reaches 500. After the number of evacuees reached 500, the growth rate of RSET gradually accelerated. This is because when there are too many evacuees, the evacuation model does not set the path capacity constraints, which does not make full use of the free space of each path, and eventually causes serious congestion on multiple paths. In addition, according to the settings of ASET, it can be judged that when the number of evacuees is 400, RSET is less than 7 minutes. When the number of evacuees is 450, RSET is greater than 7 minutes. Therefore, this paper believes that 400 evacuees is the maximum capacity of the model. This means that the model can accommodate up to 400 evacuees within the safety limit. This paper uses the maximum capacity within the safety limit, the value of RSET and EDRT, and the growth rate of RSET and EDRT to evaluate the results of the optimization of the evacuation model.

Comment 19: Please explain what is meant with class I, II, III and IV fire resistance and add some sources for the ASET values reported.

Reply: Thank the reviewer for the suggestions. In the revised manuscript, we give a more detailed explanation on page 16. (line409-428).

For fires in buildings, the available safe egress time (ASET) represents the time required for the building to reach the durability constraints. The fire resistance rating is a grading scale that measures the fire resistance of a building. It is determined by the combustion performance and fire resistance limit of the components that make up the building. The fire resistance rating of a building is one of the most basic measures in the fire protection technical measures specified in the building design fire protection code. The factors that affect the selection of fire resistance levels are: the importance of the building, the nature of use and fire hazard, the height and area of the building, the size of the fire load and other factors. The main building components of Class I fire-resistant buildings are all non-combustible. The main building components of Class II fire-resistant buildings are non-combustible except for the ceiling which is difficult to burn. The roof load-bearing components of Class III fire-resistance buildings are flammable. The fire wall of the Class â…£ fire-resistance building is non-combustible, and the rest are flame-retardant and flammable. For civil buildings and high-rise buildings with Class I and II fire resistance, ASET generally takes 5-7 minutes; for Class III and IV fire resistance ASET buildings, 2-4 minutes. The materials in cruise ships are generally non-flammable, so this article uses ASET=7minutes as a reference indicator to determine the maximum personnel capacity within the safety limits of different evacuation models. [46]. The relationship between the total number of evacuees and the evacuation time drawn according to the data in Table 4 is shown in Figure 12.

Comment 20: Please explain how the number of evacuees can be controlled in actual evacuations.

Reply: Thank the reviewer for the suggestions. In practice, positioning technology can be used to collect the number of evacuees on different paths. These data are then sent to the crew, and finally the crew will command the evacuation of different paths. In this way, the capacity of the path is controlled. In the revised manuscript, we give a more detailed explanation on page 18. (line462-465).

For the actual evacuation situation, the method of capacity constraints can be used for reference, and the evacuees can be equipped with smart bracelets, signs and crews who direct the evacuation to control the number of evacuees on each path.

Comment 21: Please explain what is meant with path equipped with capacity constraints and evacuation priority.

Reply: Thank the reviewer for the suggestions. Since our mother tongue is not English, we made a mistake in expression. In the revised manuscript, we give a more detailed explanation on page 19. (line489-491).

Therefore, this paper believes that the evacuation model with path capacity constraints can greatly improve the safety of evacuation.

Comment 22: Use numbers instead of using generic terms like “slightly”, “too large”, “optimal” etc. These terms are vague and do not indicate accurate discussion of the results and the conclusion.

Reply: Thank the reviewer for the suggestions. We have rewritten the conclusions and experimental analysis in the paper on page 19-22.

The evacuation route and initial location of all evacuees in this paper are always the same. Figure 13(a) shows that  will affect RSET according to the polyline trend, and different settings of  have different effects on RSET. When  and the number of evacuees is 850, the corresponding RSET is 290s, which is 130s smaller than the set ASET=7min. In addition, it can be seen from the Figure 13(a) that the overall increase rate of evacuation time is also the smallest, and the capacity within the maximum safety limit exceeds 850 people. Therefore, this paper believes that  is the optimal value. Figure 13(b) shows that when there is no path capacity constraints and the number of evacuees exceeds 450 people, the RSET corresponding to  begins to rise significantly, and was much larger than the RSET corresponding to . In the end, when the number of evacuees is 850, the corresponding RSET is 55005s, which is far more than 290s. Therefore, this paper believes that the evacuation model with path capacity constraints can greatly improve the safety of evacuation. (line478-491)

When, the RSET in Table 9 increases within 50s compared to the RSET in Table 7. When  and the number of evacuees is 800-850, the RSET in Table 9 is reduced by 100-250s compared to the RSET in Table 7. When the evacuees have the evacuation priority, the evacuation time will increase or decrease to different degrees at different , but the increase is not very large. This proves that the overall evacuation time has not been greatly affected after the evacuees’ evacuation priority. However, when the number of evacuees is between 450 and 850, the EDRT in Table 10 is reduced within 50s compared to the EDRT in Table 8. (line501-508)

When and number of evacuees exceeds 450, The evacuation time will be longer than the evacuation time corresponding to . when , it will greatly save evacuation time and improve system safety. In addition, it can be seen from Figures 16(b) and 17(b) that RSET and EDRT will increase slightly, when the number of evacuees is between 50 and 600 after setting the evacuation priority. However, when the number of evacuees between 650 and 850, compared with the EDRT with only path capacity constraints, the EDRT of the personnel priority evacuation model will be reduced and the corresponding RSET difference is not obvious. This proves that when too many evacuees are evacuated, the priority evacuation model can ensure the safety of evacuees in high-risk paths and shorten their evacuation time, while ensuring that the overall evacuation time is not greatly affected. (line521-532)

Based on the fire emergency, this paper proposes the priority evacuation model with path capacity constraints. The improvement of the shortest evacuation route based on the fire node, and the program design of the attributes change of evacuation network topology and the evacuees are given. The evacuation simulation was carried out based on Python 3.6. The priority evacuation model with path capacity constraints was compared with the original evacuation model. The experimental results in this paper analyze that when the number of evacuees exceeds 600, the evacuation time without path capacity constraints will exceed 2952s. After the path capacity constraints are set and the number of evacuees is 850, the maximum evacuation time is only 731s. This proves that setting the path capacity constraints ensures the safety of all overall evacuation. The difference in the maximum crowd density of the path will also affect the evacuation time. According to the experimental results, when , the RSET and EDRT are the shortest. In addition, when the number of evacuees between 650 and 850, the safety of evacuees on dangerous paths will be further guaranteed. This model can not only improve the safety of evacuees near the fire area, but also can prevent congestion in various paths and reduce the RSET.

In actual emergencies, the path capacity constraints and priority evacuation model proposed in this paper can provide guidance. When the number of evacuees is between 0 and 600, the path capacity constraints shall be set during evacuation to improve the safety of evacuees. When the number of evacuees is between 650 and 850, the priority evacuation model is adopted to ensure the safety of evacuees in local dangerous areas. In actual evacuation, measures such as smart bracelets, signs, rescuers, etc. can be used to provide path capacity constraints and the priority evacuation.

In addition, this article does not consider the dynamic spread of the fire and the dynamic change of the corresponding path dangerous degree, which will be carried out in the follow-up research.

(line538-563)

Comment 23: REST should be RSET.

Reply: Thank the reviewer for the suggestions. We modify the abbreviation errors in the full text.

Round 2

Reviewer 2 Report

The paper is well organized and structured, and most of this Reviewer’s previous comments have been addressed. Proper referencing has been added and the equations and parameters used are well explained. However, the sentence structure of the newly added paragraphs and the use of verbs and expressions which are not very clear makes it difficult for the reader to fully comprehend – this needs to be improved. The conclusion section has improved significantly but lacks flow between sentences. The gap that this paper fills must be highlighted in the abstract, introduction and conclusion sections.

Specific Comments

 Line 8

Should be “Large buildings such as shopping malls and cruise ships are crowded and complex”

Line 38 – Please clarify online simulation? Are there references available? This phrase needs grammatical revision

Line 46

BIM is already defined in line 43, no need to use the full term again.

Line 47

Please fix the sentence beginning with ‘After analyzing the IFC, connect…’ is missing: is it still Mirahadi?

Line 52

Please clarify the difference between the terms ‘scenario’ and ‘situation?’

Line 60,

HIPLA stands for hierarchical path planning algorithm and not hierarchical path planning method

Line 71

Please clarify ‘spatial innovation’

Line 100

Please clarify what is meant by ‘rationality of the model’, maybe it is meant to be ‘validation’ of the model.

Line 119

Please rephrase ‘This paper selects … and judges….’ It is preferable to  state ‘method’ rather than ‘paper’

Line 124

Please state the requirements of evacuation simulation.

Line 126

Please clarify personnel – is it the ship crew? Which criteria are used to select the escape area?

Line 136

‘…is the shortest distance both the sides of the path’, perhaps between the sides of the path?

Lines 158-162

The wording of this paragraph needs to be improved as the term “in this paper” or “this paper” is repeated multiple times. It can be mentioned once and the flow of the sentence is already showing that the whole paragraph is about the current research.

Line 176-178

Sentence is not clear

Line 199-202

I know this paragraph is added to explain what is meant by ASET has more safety margin than RSET, but the sentence itself is not properly constructed. Adding the formulas of both ASET and RSET may make things clearer for the reader to understand. Also please ensure that the definition of ASET corresponds to internationally generally accepted definitions such as those in SFPE Handbook of Fire Protection Engineering and please provide references.

Line 224-236

The English language of this paragraph is poor and need to be revised (e.g. “It can be seen”, “Fruin and Nelson and Mowrer models”, “their speed model is not perfect” etc.)

Line 234 and 236

Please use a different word rather than “synthesize” as it is not clear.

Line 392

In the Evacuee’s Movement Model code the term REST is used. I think it should be changed to RSET

Line 394

In the figure, it is not clear what is the initial node of the graph: ‘incoming time steps and evacuation routes to personnel’.

Lines 409-428

Please provide references for classifications of fire resistance

Line 422-424

Please add a reference for the ASET values obtained for the different classes

Line 521-523

The sentence is not clear and needs to be rewritten.

Line 542

It is stated that the proposed model is compared to the original model. Please add a concluding sentence to show the outcome of this comparison. (e.g. it improves a specific outcome by 10%)

Author Response

Authors Response to the Review of jmse-1467077

Dear reviewer,

Thanks very much for your careful reading and insightful comments. We are pleased to answer the suggestions and the manuscript (Manuscript No.: jmse-1467077) has also been extensively revised according to the comments, and the detailed corrections are listed below:

Reviewer: 2

Comment 1: Should be “Large buildings such as shopping malls and cruise ships are crowded and complex”. Please clarify online simulation? Are there references available? This phrase needs grammatical revision.

Reply: Thank the reviewer for the suggestions. Due to our negligence, some grammatical and inaccurate expression problems appeared. There is an error in the expression of the online simulation, and it should be an evacuation simulation. It is an offline simulation. In the revised manuscript, we have corrected on page 2 (line 8 and line 38).

Large buildings such as shopping malls and cruise ships are crowded and complex.

In evacuation simulation, the first step is to model the indoor environment of the building.

Comment 2: BIM is already defined in line 43, no need to use the full term again. Please fix the sentence beginning with ‘After analyzing the IFC, connect…’ is missing: is it still Mirahadi?

Reply: Thank the reviewer for the comments and the suggestions. In the revised manuscript, we have corrected on page 2. (line46 and line47).

Farid Mirahadi [11] uses BIM to provide facility functions and physical characteristics for other modules, and then converts BIM to Industry Foundation Classes (IFC) data structure. After analyzing the IFC, he generates a geometric network graph (GNG) by connecting the center points of different regions.

Comment 3: Please clarify the difference between the terms ‘scenario’ and ‘situation?’. HIPLA stands for hierarchical path planning algorithm and not hierarchical path planning method.

Reply: Thank the reviewer for the suggestions. The evacuation scene refers to the place where people escape, and the evacuation situation refers to different emergencies. Since our mother tongue is not English, it has caused a deviation in expression. We replaced scenario with building. In the revised manuscript, we have corrected on page 2. (line54 and line61).

For different buildings and different dangerous situations, the results of the same path planning algorithm are different[15]. For example, fires in subways and office buildings, poisonous gas leaks in chemical plants, fires or collisions in cruise ships, etc.

Ramesh et al. [18] used the hierarchical path planning algorithm (HIPLA) to solve the risk minimization problem in dynamic hazardous environments.

Comment 4: Please clarify ‘spatial innovation’. Please clarify what is meant by ‘rationality of the model’, maybe it is meant to be ‘validation’ of the model.

Reply: Thank the reviewer for the suggestions. Spatial innovation means that the micro model considers the impact of the environment on people. In the revised manuscript, we have added an explanation on page 2 (line72 and line 93).

At the micro level, continuous or discrete models can be used for simulation, focusing on pedestrian evacuation and spatial innovation. The movement of pedestrians is determined by factors such as pedestrian characteristics and surrounding environment.

In order to verify validation of the evacuation model in this paper and analyze the advantages and disadvantages of the model performance

Comment 5: Please rephrase ‘This paper selects … and judges….’ It is preferable to state ‘method’ rather than ‘paper’. Please state the requirements of evacuation simulation. Please clarify personnel – is it the ship crew? Which criteria are used to select the escape area?

Reply: Thank the reviewer for the suggestions. This requirement means that the fire source and escape exit must be set up in advance for evacuation. Persons here refer to pedestrians. In the revised manuscript, we have corrected on page 3. (line119 and line124-126).

Therefore, this method first selects the center points of these areas,

In addition, in order to study the impact of path capacity constraints and evacuation priority on evacuation, it is necessary to set up the fire location and exit as the prerequisite for evacuation simulation.

In "SAFEGUARD" project, pedestrians were designated to flee to assembly area C. Assembly area C contains the rescue materials needed for emergency evacuation, so this paper takes the node leading to assembly area C as the exit.

Comment 6: ‘…is the shortest distance both the sides of the path’, perhaps between the sides of the path? The wording of this paragraph needs to be improved as the term “in this paper” or “this paper” is repeated multiple times. It can be mentioned once and the flow of the sentence is already showing that the whole paragraph is about the current research. Line 176-178 Sentence is not clear

Reply: Thank the reviewer for the suggestions. In the revised manuscript, we have corrected on page 3 (line137) and page 5 (line160-162 and line177-179)

represents the width of the path, and its value is the shortest distance between the sides of the path.

This paper focuses on the impact of setting path capacity constraints and evacuation priority on the evacuation results. The method of determining the risk level of a path can be simplified. Therefore, the risk level of all paths is judged by the distance from all areas to the fire area.

When constructing the network topology for some other large buildings, the extended attributes of this paper can be used as a reference, such as crowd density, the shortest route, the types of nodes and risk level of paths.

Comment 7: I know this paragraph is added to explain what is meant by ASET has more safety margin than RSET, but the sentence itself is not properly constructed. Adding the formulas of both ASET and RSET may make things clearer for the reader to understand. Also please ensure that the definition of ASET corresponds to internationally generally accepted definitions such as those in SFPE Handbook of Fire Protection Engineering and please provide references.

Reply: Thank the reviewer for the suggestions. We have added formulas and references for RSET and ASET. In the revised manuscript, we have corrected on page 3. (line198-208).

Evaluation of evacuation results and evacuation models can use evacuation time as the indicator. There are two types of evacuation time, available safe egress time (ASET) and required safe egress time (RSET) [36, 37]. The different types of times included in ASET and RSET are shown in formulas (5) and (6).

                             (5)

                                (6)

In formula (7) and (8),  represents the time required for the pedestrian to detect the fire, call the police, and then final response. represents the time required for the actual movement of the pedestrian.  represents the time for the building to reach its fire-resistance limitation minus RSET [38].

[38] Philip J.; DiNenoo, P.E. SFPE Handbook of Fire Protection Engineering, 3rd ed.; Publisher: National Fire Protection Association, America, 2002; pp. 1–65.

Comment 8: The English language of this paragraph is poor and need to be revised (e.g. “It can be seen”, “Fruin and Nelson and Mowrer models”, “their speed model is not perfect” etc.)

Please use a different word rather than “synthesize” as it is not clear.

Reply: Thank the reviewer for the suggestions. In the revised manuscript, we have corrected on page 7-8. (line227-237).

Figures 3, 4, and 5 show the evacuees’ moving speed on the stairs and the straight path, respectively. Since the speed models proposed by scholars are based on the evacuation experimental data in different buildings, it has led to large differences among their models. It can be seen from Figure 3 that when the crowd density exceeds 4 people/m², the Fruin model and the Nelson-Mowrer model will no longer be applicable. In practice, the high crowd density will not cause evacuees’ moving speed to be 0. And when the crowd density is close to 0 people/m², there is a big difference among the scholar models. Therefore, scholars’ speed model need be improved. In order to reduce the influence of the differences and improve the evacuees’ speed model, this paper integrates the speed models of various scholars to make the new model as closer to the actual situation.

Comment 9: In the Evacuee’s Movement Model code the term REST is used. I think it should be changed to RSET. In the figure, it is not clear what is the initial node of the graph: ‘incoming time steps and evacuation routes to personnel’.

Reply: Thank the reviewer for the suggestions. We have corrected the corresponding spelling errors in the pseudo code and redrawn the flow chart. In the revised manuscript, we have corrected on page 11. (line302-319).

Comment 10: Please provide references for classifications of fire resistance. Please add a reference for the ASET values obtained for the different classes.

Reply: Thank the reviewer for the suggestions. In the revised manuscript, we added references for classifications of fire resistance and the ASET values obtained for the different classes on page 16 (line418-428).

The main building components of Class I fire-resistant buildings are all non-combustible. The main building components of Class II fire-resistant buildings are non-combustible except for the ceiling which is difficult to burn. The roof load-bearing components of Class III fire-resistance buildings are flammable. The fire wall of the Class â…£ fire-resistance building is non-combustible, and the rest are flame-retardant and flammable [47]. For civil buildings and high-rise buildings with Class I and II fire resistance, ASET generally takes 5-7 minutes. For civil buildings and high-rise buildings with Class III and IV fire-resistance, ASET generally takes 2-4 minutes [47]. The materials in cruise ships are generally non-flammable, so this paper uses ASET=7minutes as a reference indicator to determine the maximum capacity within the safety limitation of different evacuation models [48].

[47] Tianjin Fire Research Institute of M.E.M. Code for design of building fire protection, GB50016-2014.; Publisher: Guohong Building, No.11, Muxidi Beili, Xicheng District, Beijing, China, 2014; pp. 57–75

Comment 11: (Line521-523) The sentence is not clear and needs to be rewritten.

Reply: Thank the reviewer for the suggestions. In the revised manuscript, we have corrected on page 22 (line518-521).

Figure 15 shows the impact of the model after adding the evacuees’ priority on RSET and EDRT. As the number of evacuees increases, the RSET and EDRT tend to increase. In addition, RSET and EDRT are also related to . When number of evacuees exceeds 450, the RSET with  will be longer than the RSET with .

Comment 12: It is stated that the proposed model is compared to the original model. Please add a concluding sentence to show the outcome of this comparison. (e.g. it improves a specific outcome by 10%).

Reply: Thank the reviewer for the suggestions. In the revised manuscript, we have corrected on page 22 (line540-542).

Based on the obtained simulation data, it is found that when a large number of evacuees are evacuated inside the cruise ship, the priority evacuation model with path capacity constraints reduces RSET to less than 600s.

In addition, we have rewritten the sentences in the simulation results analysis and conclusion part. Finally, thank you very much for your valuable comments on our paper.
